# Remyelination alters the pattern of myelin in the cerebral cortex

**Jennifer Orthmann-Murphy[1,2†]\*, Cody L Call[1†], Gian C Molina-Castro[1], Yu Chen Hsieh[1], Matthew N Rasband[3]\*, Peter A Calabresi[4], Dwight E Bergles[1,5]\***

[1]The Solomon Snyder Department of Neuroscience, Johns Hopkins University, Baltimore, United States; [2]Department of Neurology, Perelman School of Medicine, University of Pennsylvania, Philadelphia, United States; [3]Department of Neuroscience, Baylor College of Medicine, One Baylor Plaza, Houston, United States; [4]Department of Neurology Johns Hopkins University, Baltimore, United States; [5]Johns Hopkins University Kavli Neuroscience Discovery Institute, Baltimore, United States

**Abstract** Destruction of oligodendrocytes and myelin sheaths in cortical gray matter profoundly alters neural activity and is associated with cognitive disability in multiple sclerosis (MS). Myelin can be restored by regenerating oligodendrocytes from resident progenitors; however, it is not known whether regeneration restores the complex myelination patterns in cortical circuits. Here, we performed time lapse in vivo two photon imaging in somatosensory cortex of adult mice to define the kinetics and specificity of myelin regeneration after acute oligodendrocyte ablation. These longitudinal studies revealed that the pattern of myelination in cortex changed dramatically after regeneration, as new oligodendrocytes were formed in different locations and new sheaths were often established along axon segments previously lacking myelin. Despite the dramatic increase in axonal territory available, oligodendrogenesis was persistently impaired in deeper cortical layers that experienced higher gliosis. Repeated reorganization of myelin patterns in MS may alter circuit function and contribute to cognitive decline.

**\*For correspondence:**
Jennifer.Orthmann-Murphy@
pennmedicine.upenn.edu (JO-M);
rasband@bcm.edu (MNR);
dbergles@jhmi.edu (DEB)

[†]These authors contributed equally to this work

## Introduction

Oligodendrocytes form concentric sheets of membrane around axons that enhance the speed of action potential propagation, provide metabolic support and control neuronal excitability through ion homeostasis. Consequently, loss of oligodendrocytes and myelin can alter the firing behavior of neurons and impair their survival, leading to profound disability in diseases such as multiple sclerosis (MS), in which the immune system inappropriately targets myelin for destruction. In both relapsing-remitting forms of MS and the cuprizone model of demyelination in mouse, the CNS is capable of spontaneous remyelination through mobilization of oligodendrocyte precursor cells (OPCs) (*Baxi et al., 2017*; *Chang et al., 2000*; *Chang et al., 2012*), which remain abundant in both gray and white matter throughout adulthood (*Dimou et al., 2008*; *Hughes et al., 2013*; *Young et al., 2013*). The highly ordered structure of myelin in white matter tracts has enabled in vivo longitudinal tracking of inflammatory demyelinating lesions using magnetic resonance imaging (MRI); however, due to the low spatial resolution of standard MRI sequences (*Oh et al., 2019*) and the indirect nature of MRI methods used to assess the integrity of myelin (*Walhovd et al., 2014*), our knowledge about the dynamics of OPC recruitment, oligodendrogenesis and remyelination within specific neural circuits remains limited.

In vivo studies of remyelination have focused primarily on white matter, where assessments of myelin are aided by the high density and symmetrical alignment of myelin sheaths; however, postmortem histological analysis (*Kidd et al., 1999*; *Kutzelnigg et al., 2005*; *Lucchinetti et al., 2011*;

*Peterson et al., 2001*) and new in vivo MRI and PET imaging methods (*Beck et al., 2018*; *Filippi et al., 2014*; *Herranz et al., 2019*; *Kilsdonk et al., 2013*; *Magliozzi et al., 2018*) indicate that demyelination is also prevalent in cortical gray matter of MS patients. Cortical lesion load correlates with signs of physical and cognitive disability, including cognitive impairment, fatigue and memory loss (*Calabrese et al., 2012*; *Nielsen et al., 2013*). Nevertheless, much less is known about the capacity for repair of myelin in cortical circuits. Defining how gray matter lesions are resolved in vivo is critical for both MS prognosis and the development of new therapies to promote remyelination.

Unlike white matter, myelination patterns in the cerebral cortex are highly variable, with sheath content varying between cortical regions, among different types of neurons and even along the length of individual axons (*Micheva et al., 2016*; *Tomassy et al., 2014*). Despite this evidence of discontinuous myelination, recent in vivo imaging studies indicate that both oligodendrocytes and individual myelin sheaths are remarkably stable in the adult brain (*Hill et al., 2018*; *Hughes et al., 2018*), suggesting that maintaining precise sheath placement is important for cortical function. However, the complex arrangement of cortical myelin may present significant challenges for repair and it is unknown whether precise myelination patterns in the cortex are restored following a demyelinating event.

In vivo two photon fluorescence microscopy allows visualization of oligodendrogenesis and myelin sheath formation in mammalian circuits at high resolution, providing the means to define both the dynamics and specificity of regeneration (*Hughes et al., 2018*). Here, we used this high-resolution imaging method to define the extent of oligodendrocyte regeneration and the specificity of myelin replacement within the adult somatosensory cortex after demyelination. Unexpectedly, our studies indicate that oligodendrocytes are regenerated in locations distinct from those occupied before injury. Despite the additional available axonal territory for myelination, regenerated oligodendrocytes had a similar size and structure; as a result, only a fraction of prior sheaths were replaced and many new sheaths were formed on previously unmyelinated regions of axons. Conversely, in regions of high territory overlap between original and regenerated cells, new oligodendrocytes often formed sheaths along the same segment of specific axons, demonstrating that precise repair is possible. Together, these in vivo findings indicate that regeneration of oligodendrocytes in the cortex creates a new pattern of myelination, with important implications for the restoration of sensory processing and cognition.

## Results

### Inefficient regeneration of oligodendrocytes in cortical gray matter

To define the dynamics of oligodendrocyte regeneration and axonal remyelination in the cerebral cortex, we performed longitudinal two photon imaging through a cranial window placed over the barrel field of the somatosensory cortex in transgenic mice that express EGFP under control of the promoter/enhancer for myelin-associated oligodendrocyte basic protein (*Mobp-EGFP*) mice (*Hughes et al., 2018*; *Figure 1A*). In these mice, complete oligodendrocyte morphologies could be resolved in vivo, including cytoplasmic processes and individual myelin internodes within the upper layers of the cortex. In these regions, oligodendrocytes ensheath a select group of axons, including long-range axonal projections oriented horizontally to the pia (*Bock et al., 2011*; *Figure 1B*), and in deeper layers, vertically-oriented axons belonging to both local cortical neurons and long-range projections (*Figure 1A,C*). To induce demyelination, young adult *Mobp-EGFP* mice (age 8–12 weeks) were fed chow mixed with 0.2% cuprizone, a copper chelator that induces robust fragmentation and apoptosis of oligodendrocytes (*Vega-Riquer et al., 2019*; *Figure 1—figure supplement 1*), and multiple volumes (425 µm x 425 µm x 550 µm) corresponding to layers I–IV were imaged repeatedly prior to injury, during demyelination and through recovery for up to 12 weeks (*Figure 1D*; *Video 1*).

Oligodendrocytes and individual myelin sheaths are extraordinarily stable in the adult brain (*Hill et al., 2018*; *Hughes et al., 2018*; *Yeung et al., 2014*); however, the amount of myelin within these circuits is not static, as new oligodendrocytes continue to be generated in the adult CNS, each of which produces dozens of sheaths (*Hughes et al., 2018*; *Kang et al., 2010*; *Xiao et al., 2016*). This phenomenon was visible during in vivo imaging in *Mobp-EGFP* mice, as new EGFP-expressing (EGFP+) oligodendrocytes appeared within the imaging field (*Figure 1E,F,H*), continuously adding

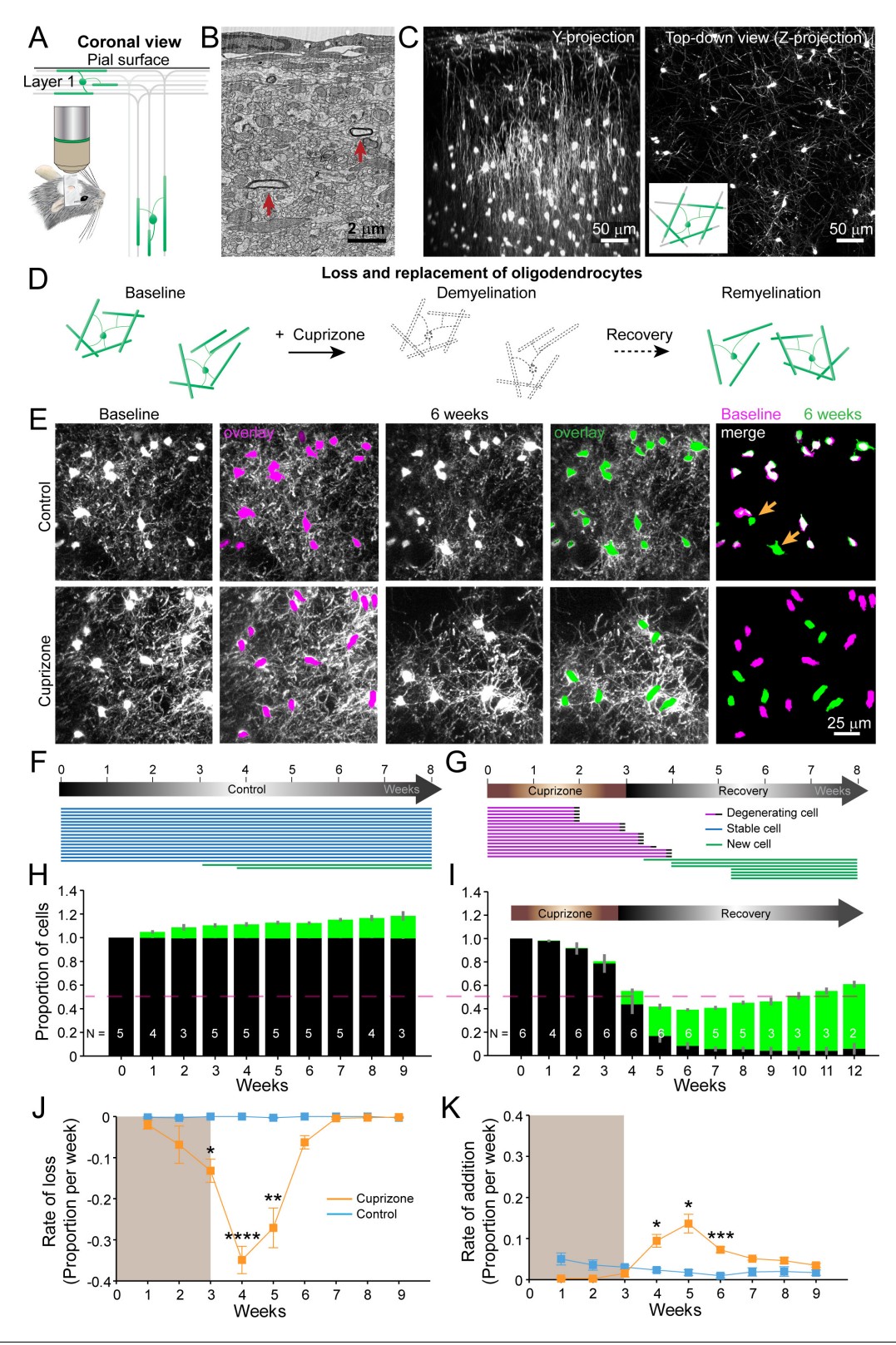

**Figure 1.** An in vivo platform to monitor loss and replacement of oligodendrocytes in the cerebral cortex. (**A**) In vivo two photon microscopy through chronic cranial windows over the somatosensory cortex of *Mobp-EGFP* mice (coronal view), showing myelinated fibers in cortical layer I parallel to pial surface and in deeper layers oriented perpendicularly. (**B**) Electron micrograph reconstruction of adult mouse visual cortex (from *Bock et al., 2011*) illustrating low density of myelinated fibers (arrows) in the upper layers of cortex. (**C**) Maximum intensity *y*-projection (coronal view, 425 µm x 150 µm x

*Figure 1 continued on next page*

*Figure 1 continued*

550 µm) and z-projection (top-down view, 425 µm x 425 µm x 100 µm) example regions from *Mobp-EGFP* mice with chronic cranial windows. (**D**) Schematic illustrating longitudinal course of loss (demyelination) and replacement (remyelination) of cortical oligodendrocytes. (**E**) Examples of maximum intensity projection images of the same region (156 µm x 156 µm x 84 µm) imaged repeatedly from an adult sham- (control, top row) or a cuprizone-treated (bottom row) mouse are shown with overlay of cell bodies from baseline (magenta) and after 6 weeks (green). Merge of baseline and 6 week overlays show where new cells are added to the region (arrows). (**F-G**) Individual cells (represented by magenta, blue or green lines) were tracked longitudinally in somatosensory cortex from mice fed control (F; from region in top row of E) or cuprizone diet (G; from region in bottom row of E). (**H-K**) The same cortical volume (425 µm x 425 µm x 300 µm) was imaged repeatedly in mice given either control or cuprizone diet, and individual cells present at baseline (black) or formed at later time points (green) were tracked over time. Shown are the average cell counts depicted as a proportion of baseline number of cells, (H, N = 5 control mice; I, N = 6 cuprizone mice, I; number of mice imaged at each time point indicated). (**J-K**) The average rate of loss (**J**) or addition (**K**) of oligodendrocytes per week in control-treated (blue) v. cuprizone-treated mice (orange) relative to the baseline population of oligodendrocytes. Treatment with sham or cuprizone-supplemented chow denoted by shaded background. In cuprizone-treated mice, there was a higher rate of oligodendrocyte loss over weeks 3–5 and addition of new cells between 4–6 weeks compared to control. Data is presented as means with standard error of the mean bars. See *Supplementary file 1* for statistical tests and significance level not otherwise noted. The online version of this article includes the following figure supplement(s) for figure 1:

**Figure supplement 1.** Degeneration of oligodendrocytes in cuprizone-treated *Mobp-EGFP* mice.
**Figure supplement 2.** Preservation of cortical axons following cuprizone treatment.
**Figure supplement 3.** Non-uniform distribution of myelin in the adult rodent cortex.

to the baseline oligodendrocyte population (*Figure 1H*; *Video 2*). When mice were fed cuprizone for three weeks, >90% of the baseline population of oligodendrocytes within the upper layers of cortex degenerated (94.2 ± 0.05%; N = 6 mice, mean ± SEM) (*Figure 1E,G,I*; *Video 3*), with most cell loss occurring after cessation of cuprizone exposure (*Figure 1I,J*). New oligodendrocytes initially appeared rapidly during the recovery phase (*Figure 1K*); however, this burst of oligodendrogenesis was not sustained, and as a consequence, only about half of the oligodendrocytes (55.2 ± 0.03%) were replaced after nine weeks of recovery. Extrapolating from the rate of addition from the last recovery time-point (3.5 ± 0.5% per week, *Figure 1K*), it would take approximately three additional months (~12.8 weeks) to achieve the density of oligodendrocytes prior to cuprizone. Oligodendrocyte density increases with age and is ~two fold greater in middle-aged cortices (*Hughes et al., 2018*). To account for the addition of oligodendrocytes that normally occurs, these young mice would be >13 months old by the time they achieved full recovery, assuming that the rate of addition is sustained. However, as oligodendrogenesis declines with age in both control and cuprizone-treated mice (*Figure 1K*), cuprizone-treated mice may never reach a normal oligodendrocyte density after demyelination. These results indicate that although a prominent regenerative response is initiated in cortical gray matter, oligodendrocyte regeneration is much slower (*Baxi et al., 2017*) and less complete than in white matter (*Baxi et al., 2017*; *Gudi et al., 2009*; *Jeffery and Blakemore, 1995*; *Matsushima and Morell, 2001*).

Axonal degeneration has been observed in the experimental allergic encephalomyelitis (EAE) demyelinating model (*Mei et al., 2016*), and there is evidence of cortical axon pathology in the corpus callosum after prolonged exposure to cuprizone (*Gudi et al., 2009*). Axon loss could contribute to the decrease in remyelination

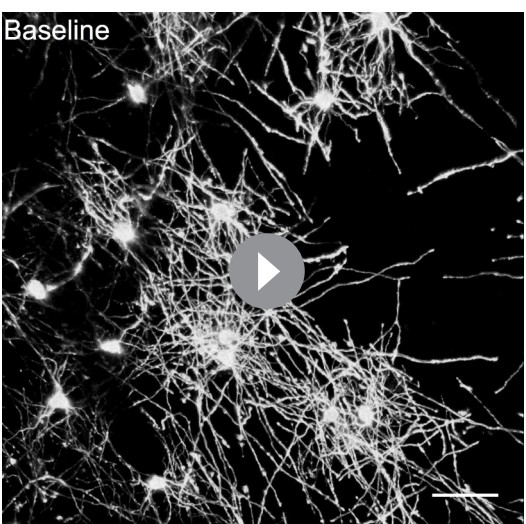

**Video 1.** Loss and replacement of oligodendrocytes. Longitudinal in vivo imaging of demyelination and remyelination. This is a 392 µm x 392 µm x 100 µm volume shown as a maximum intensity projection that was repeatedly imaged through a chronic cranial window over the somatosensory cortex in an adult *Mobp-EGFP* mouse, at baseline, over 3 weeks of cuprizone administration, and then through 5 weeks of recovery. Scale bar is 50 µm.
https://elifesciences.org/articles/56621#video1

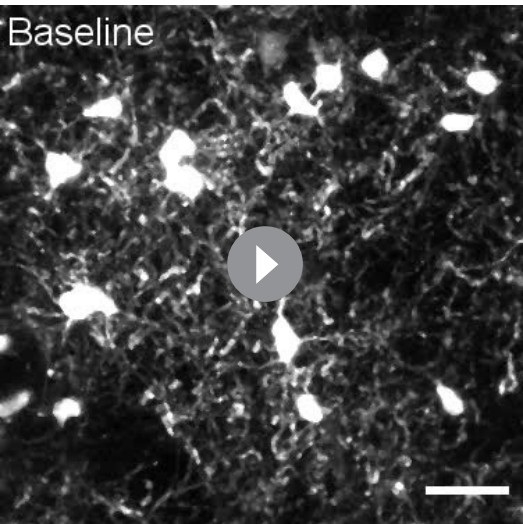

**Video 2.** New oligodendrocytes are added in the upper cortical layers in adult mice. Longitudinal imaging of an adult *Mobp-EGFP* mouse with a chronic cranial window fed sham diet. Region corresponds to images shown in *Figure 1E*, top row. Scale bar is 25 µm.
https://elifesciences.org/articles/56621#video2

**Video 3.** Oligodendrocytes are lost and new cells appear after cuprizone-treatment. Longitudinal imaging of an adult *Mobp-EGFP* mouse with a chronic cranial window fed 3 weeks of a cuprizone-supplemented diet followed through 5 weeks of recovery. Region corresponds to images shown in *Figure 1E*, bottom row. Scale bar is 25 µm.
https://elifesciences.org/articles/56621#video3

observed; however, this pathology would have to be extensive to account for the ~45% decrease in oligodendrocyte regeneration. To determine whether there is marked loss of axons in this 3-week cuprizone protocol, we performed post-hoc immunostaining for neurofilament-L, which predominantly labels axons, on sections of somatosensory cortex from control and cuprizone-treated mice after five weeks recovery (*Figure 1—figure supplement 2*). These sections exhibited comparable neurofilament-L immunoreactivity, suggesting that cuprizone-treated animals did not experience extensive axonal loss and therefore is unlikely to be responsible for impaired oligodendrocyte regeneration in the cortex.

## Layer specific differences in cortical remyelination

The cerebral cortex is a highly layered structure, in which genetically and morphologically distinct neurons form specialized subnetworks (*Lodato and Arlotta, 2015*), raising the possibility that they may adopt different myelination patterns to optimize circuit capabilities (*Micheva et al., 2016*; *Stedehouder et al., 2017*; *Tomassy et al., 2014*). Indeed, myelination patterns are highly non-uniform across the cortex (*Figure 1—figure supplement 3*) and both oligodendrocyte density and myelin content increase with depth from the cortical surface (*Hughes et al., 2018*; *Figure 1C*). However, it is not known whether the capacity for myelin repair is comparable between cortical layers. To examine depth-dependent changes in oligodendrogenesis, we subdivided imaging volumes into three 100 µm zones starting from the pial surface (*Figure 2A–C*). In control mice, the proportional rates of oligodendrocyte addition were similar between the top zone (0–100 µm) and bottom zone (200–300 µm) (top: 2.62 ± 0.37%; bottom: 2.93 ± 0.26% per week), despite their dramatically different oligodendrocyte densities (0–100 µm: 17 ± 2 cells; 200–300 µm: 79 ± 10 cells, N = 11 mice @ baseline) (*Figure 2D,E*); the only exception was for time points when no cells were incorporated into the top zone (*Figure 2E*, p=0.0036 @ 5 weeks, N-way ANOVA with Bonferroni correction for multiple comparisons). These findings suggest that oligodendrocyte enrichment in deeper layers occurs early in development, but then proceeds at similar proportional rates across cortical layers in adulthood.

In an efficient regenerative process, cell generation would be matched to cell loss; however, we found that oligodendrocyte regeneration was not proportional to their original density.

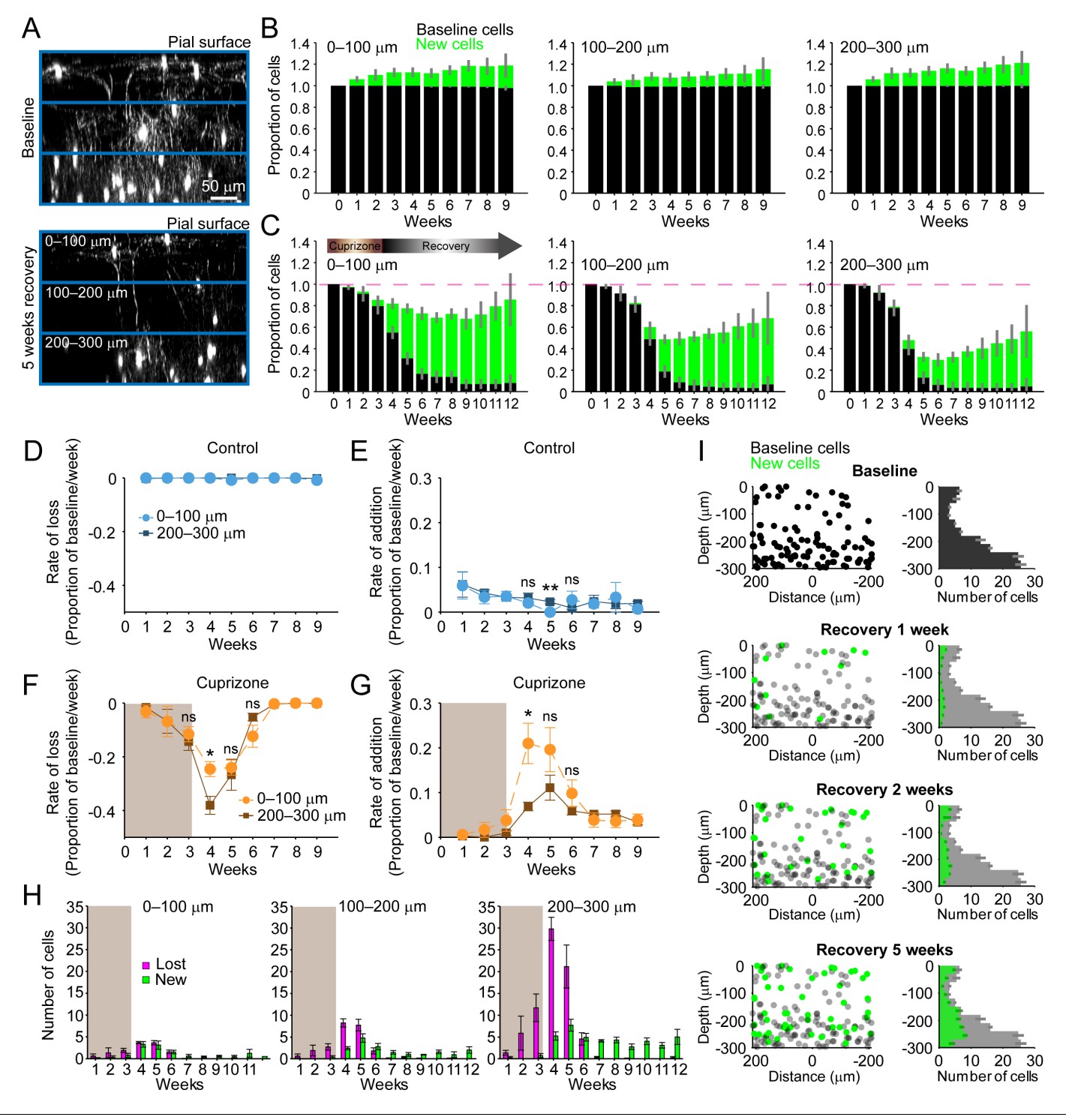

**Figure 2.** Regeneration of oligodendrocytes declines with cortical depth. (**A**) Fate of individual oligodendrocytes over time were determined within the same cortical volume (425 μm x 425 μm x 300 μm) that was divided into 0–100, 100–200, or 200–300 μm zones. Images show maximum intensity Y-projection (400 μm x 118 μm x 300 μm) at baseline (top) and later at 5 weeks recovery (bottom). B-C: Histograms showing fate of existing (black) or newly generated oligodendrocytes (green) in mice fed normal (**B**) or cuprizone-supplemented diet (**C**). Average cell counts per volume depicted as proportion of baseline cells (B: N = 5 control mice; C: N = 6 cuprizone-treated mice; same number of mice imaged at each time point as shown in *Figure 1H,I*). (**D-G**) Rate of cell loss (**D, F**) or addition (**E,G**) relative to baseline oligodendrocytes in each zone depicted for each imaging time-point as a function of cortical depth (0–100 v. 200–300 μm zones), over 9 weeks of imaging in control (**D–E**) or cuprizone-treated (**F–G**) mice. Treatment with cuprizone denoted by shaded background (**F–G**). Cells are rarely lost in control regions (**D**), and the rate of cell addition in the 0–100 (light blue circles)

*Figure 2 continued on next page*

*Figure 2 continued*

v. 200–300 µm (dark blue squares) zones are similar (except @ week 5). In the bottom 200–300 µm zone (F-G, in cuprizone-treated mice (brown squares), the rate of oligodendrocyte loss (F) is significantly greater at week 4 relative to the top (0–100 µm, orange circles) zone, and the rate of addition is significantly lower (G). H: Mean number of oligodendrocytes lost (magenta) and added (green) at each imaging time-point, between 0–100 µm and 200–300 µm zones. I: Distribution of baseline (gray) and new (green) oligodendrocyte cell bodies within one volume (left-side panels; 425 µm x 425 µm x 300 µm) and for all regions (right-side panels; mean values from N = 6 mice) at recovery weeks 1, 2 and 5 (corresponding to weeks 4, 5, and 8 of imaging weeks from A-H). Mean values depicted with error bars as standard error of the mean. See *Supplementary file 1* for statistical tests and significance level for each comparison.

The online version of this article includes the following figure supplement(s) for figure 2:

**Figure supplement 1.** Distribution of astrocytes and oligodendrocyte precursor cells over the course of cuprizone treatment and recovery.

Oligodendrocyte recovery was highly efficient in the top 100 µm zone, reaching $85.2 \pm 0.17\%$ of the baseline oligodendrocyte population nine weeks after cuprizone, but only $55.5 \pm 0.11\%$ of baseline after 9 weeks in the bottom 100 µm zone (*Figure 2C*). The peak of this oligodendrocyte loss and replacement occurred during the first 2 weeks of recovery post-cuprizone (*Figure 1J,K*, *Figure 2F, G*). During this period, there was both a proportionally higher rate of cell loss and lower rate of cell replacement in the bottom 100 µm zone compared to the upper zone (*Figure 2F,G*) (loss @ week 4, p=0.044; addition @ week 4, p=0.036, N-way ANOVA with Bonferroni correction for multiple comparisons), indicating that the ability to balance cell loss with replacement declines with depth during this initial period. Although proportionally lower, more oligodendrocytes were generated per week in deeper layers (*Figure 2H*, green bars) (0–100 µm: 0.5–3.3 cells/week; 200–300 µm: 2.8–7.7 cells/week, $p=5.82 \times 10^{-4}$ @ week 4, p=0.046 @ week 5, N-way ANOVA with Bonferroni correction for multiple comparisons), but this enhanced oligodendrogenesis was not sufficient to compensate for the greater cell loss (*Figure 2H*, magenta bars). This relative lag in regeneration is clearly visible in maps indicating where newly generated oligodendrocytes were formed with regard to depth and their corresponding density histograms (*Figure 2I*, green circles and bars). This analysis highlights that the absolute number of oligodendrocytes integrated was remarkably similar in the top and bottom zones during the first few weeks of recovery, suggesting that there may be factors that suppress regeneration of needed oligodendrocytes in deeper cortical layers.

A decrease in the pool of progenitors and the formation of reactive astrocytes are potential candidates to impair oligodendrocyte regeneration in deeper cortical layers. Although OPCs are slightly less abundant in the deeper layers compared to the surface 100 µm (*Figure 2—figure supplement 1A,C,D*) ($p=2.084 \times 10^{-13}$, N-way ANOVA, by depth), there was no evidence of persistent OPC depletion during recovery (*Figure 2—figure supplement 1D,F*) (p=0.086, Kruskal-Wallis one-way ANOVA). Reactive astrocytes, a known pathological feature of both the cuprizone model and cortical demyelinating lesions (*Chang et al., 2012*; *Skripuletz et al., 2008*), can impair OPC differentiation by secreting cytokines (*Kirby et al., 2019*; *Su et al., 2011*; *Zhang et al., 2010*), but whether reactive astrocytes impair recovery differently as a function of cortical depth is unknown. GFAP+ astrocytes are relatively rare in the deeper (200–500 µm) regions of cortex in naive mice; however, following cuprizone-treatment, their number increased nearly 10-fold ($p=5.86 \times 10^{-16}$, N-way ANOVA, by time-point). This enhanced GFAP expression remained elevated throughout the recovery period (*Figure 2—figure supplement 1G*) (p=0.006, Kruskal-Wallis one-way ANOVA, with Fisher's correction for multiple comparisons) and they retained a reactive morphology, even after 5 weeks of recovery (*Figure 2—figure supplement 1A,B,E,G*). In contrast, astrocytes in the surface 100 µm, while exhibiting higher GFAP immunoreactivity at baseline, exhibited only a transient increase that was not sustained past 2 weeks of recovery (*Figure 2—figure supplement 1E*). These findings highlight that the gliotic response to demyelination varies in magnitude and duration across the cortex, which may impair the recovery of gray matter regions with higher myelin content.

## Regeneration alters the pattern of myelination in the cortex

Myelination patterns are distinct among different neuron classes in the cortex (*Micheva et al., 2016*; *Stedehouder et al., 2017*; *Tomassy et al., 2014*), and can be discontinuous, in which myelin segments along individual axons are separated by long regions of bare axon (*Tomassy et al., 2014*). Even these isolated myelin segments are highly stable (*Hill et al., 2018*; *Hughes et al., 2018*), suggesting that preservation of these patterns is important to support cortical function, and therefore

that recreation of these patterns should be a goal of the regenerative process. We hypothesized that given the sparseness of myelination in the cortex, new oligodendrocytes generated after demyelination should be formed in locations very close to the original population. To explore the spatial aspects of oligodendrocyte replacement, we mapped the distribution of oligodendrocytes within 425 µm x 425 µm x 300 µm volumes in 3-D prior to and after recovery from cuprizone, as well as the distribution of new cells generated in control mice (*Figure 3A,B*, green circles; *Video 4*). Unexpectedly, these maps revealed that the cell bodies of regenerated oligodendrocytes occupied positions distinct from the original cells (see also *Figure 1E*). To quantify displacement of newly formed oligodendrocytes from their original locations, we calculated the Euclidean distance from each oligodendrocyte present at 5 weeks of recovery (8 weeks in control) to the location of the nearest oligodendrocyte present at the baseline time-point (*Figure 3C*). Some movement (~10 µm) was experienced by stable cells over this period ('Self-self' displacements), whether from movement (wobble) that arises through actual cell displacement or through shifts in registration over eight weeks of imaging, providing a reference to the movement that would be expected in the absence of loss and regeneration. Addition of new oligodendrocytes to the cortex over this period did not significantly alter the nearest neighbor distance (All cells vs. Self-self, 0–300 µm: p=0.284, N-way ANOVA with Bonferroni correction for multiple comparisons); however, the absolute distance between oligodendrocytes increased slightly during this time, because newly generated cells (not present at baseline) were compared to the location of other oligodendrocytes, which were often further away than the movement experienced by stable cells (*Figure 3C*, Self-self distance). In contrast, regenerated oligodendrocytes added to the cortex of cuprizone-treated mice were significantly displaced relative to oligodendrocyte positions prior to degeneration (*Figure 3C*, All Cells vs. Self-self) (p=8.95×10⁻⁷, N-way ANOVA with Bonferroni correction for multiple comparisons). This apparent rearrangement of oligodendrocyte position was not due to differences in image quality or registration across the time series, or to changes in the structure of the tissue due to cuprizone exposure, as oligodendrocytes that survived in cuprizone exhibited the same average displacement as cells in control mice over the course of 8 weeks of imaging (*Figure 3C*, Self-self, Control vs. Regenerated, p>0.05, N-way ANOVA with Bonferroni correction for multiple comparisons). This increase in displacement was also not due to incomplete recovery from demyelination, as displacement in the upper zone (0–100 µm) was greater than the average displacement across the whole volume (0–300 µm), despite the proportionally greater regeneration in layer I (*Figure 2C*). These results demonstrate that regenerated oligodendrocytes occupy locations within the parenchyma that are distinct from those present at baseline, and therefore may establish a new pattern of myelination.

The new locations of oligodendrocytes after a demyelinating event may not preclude these cells from myelinating the same axonal segments, as cortical oligodendrocytes can extend long cytoplasmic processes to form sheaths distant from the cell body (*Chong et al., 2012*; *Murtie et al., 2007*). To determine whether regenerated oligodendrocytes restore the pattern of myelination by extending longer processes, we reconstructed their complete morphologies and compared these to oligodendrocytes generated at the same age in control mice. For new oligodendrocytes that appeared in layer I in control (10 cells, N = 3 mice) or cuprizone-treated mice (9 cells, N = 3 mice), each process extending from the cell body and every myelin sheath connected to these processes were traced when the cells first appeared in the imaging volume (*Figure 3D*), and for every 2–3 days for up to 14 days. In both control and cuprizone-treated mice, newly generated oligodendrocytes exhibited an initial period of refinement after first appearance (governed by the onset of *Mobp* promoter activity and EGFP accumulation in the cytoplasm), characterized by sheath extensions and retractions (*Figure 3—figure supplement 1A–D*), pruning of entire myelin sheaths and removal of cytoplasmic processes (*Figure 3—figure supplement 1E–H*), before reaching a stable morphology (*Figure 3—figure supplement 1I,J*). The final position of the cytoplasmic process along the length of the sheaths (the likely point of sheath initiation) was randomly distributed along the length of the internode (*Figure 3—figure supplement 1K,L*). Notably, the initial dynamics of these newly formed oligodendrocytes are remarkably similar to those described in the developing zebrafish spinal cord (*Auer et al., 2018*; *Czopka et al., 2013*), suggesting that the maturation sequence of oligodendrocytes is both highly conserved and cell intrinsic.

This time-lapse analysis revealed that the morphological plasticity of newly formed oligodendrocytes lasts for more than 1 week in the cortex (*Figure 3—figure supplement 1I,J*); therefore, comparisons between control and regenerated oligodendrocytes were made from cells 12–14 days after

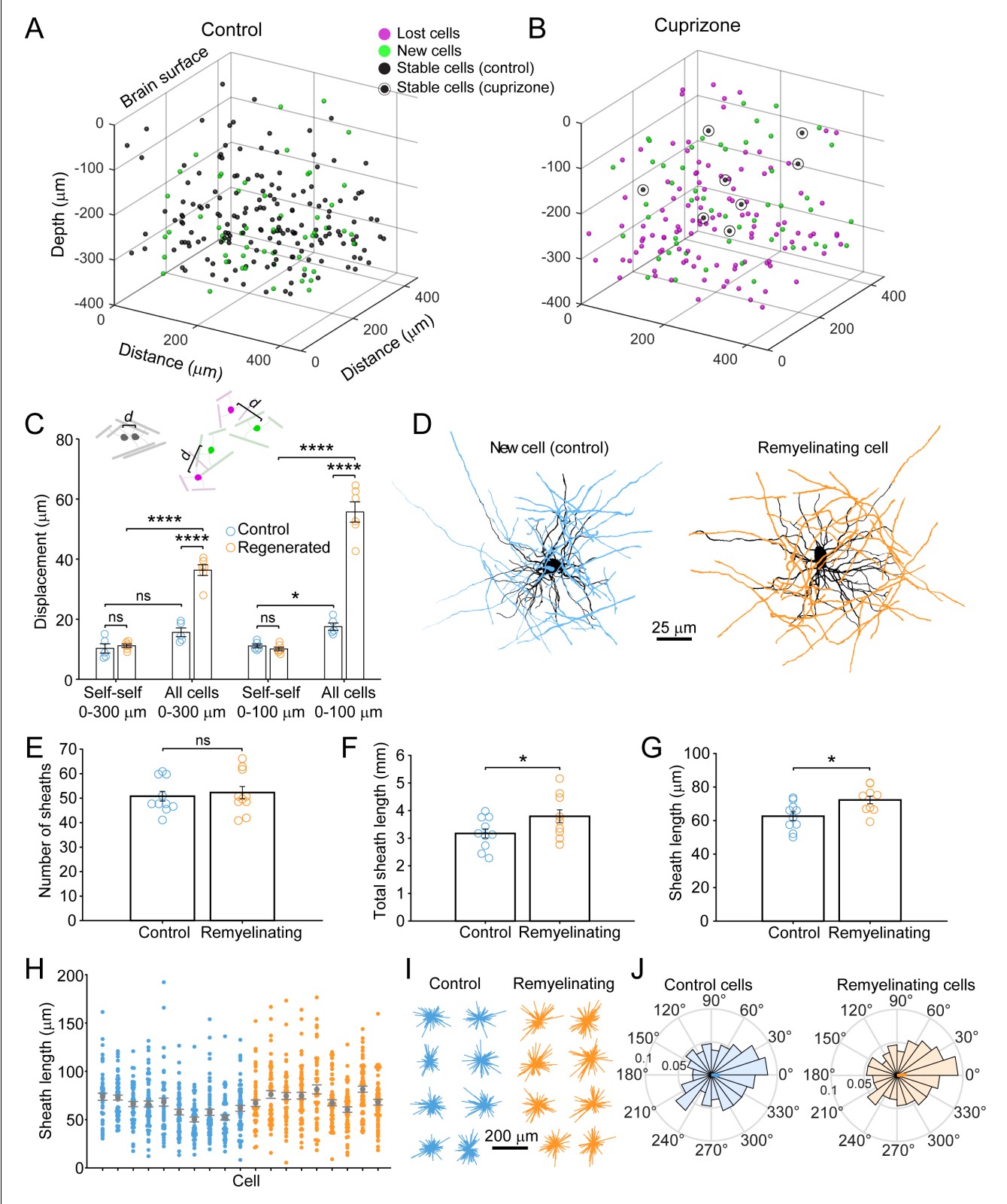

**Figure 3.** Remyelinating oligodendrocytes appear in novel locations but exhibit normal morphological characteristics. (A-B) Location of oligodendrocyte cell bodies within the same cortical volume (425 μm x 425 μm x 300 μm) at baseline and after 8 weeks of two photon in vivo imaging are plotted and overlaid in 3-D for control (A) and cuprizone-treated (B) cortex. Cell fate is designated as stable (black), lost (magenta), or new (green) (see *Video 4*). (C) Histogram showing average displacement (Euclidean distance) of regenerated oligodendrocytes at 5 weeks recovery relative to the

*Figure 3 continued on next page*

*Figure 3 continued*

nearest oligodendrocyte at baseline. *Self-self* illustrates the minor movement of cells that survived over the entire 8 weeks (schematized above by two gray oligodendrocytes displaced by small distance d). *All cells* indicates the nearest neighbor distance between regenerated and baseline cells for cuprizone-treated mice (schematized above by the green and magenta oligodendrocytes displaced by larger distance d), and for both stable and newly generated oligodendrocytes in Control. ns, not significant. (D) Examples of maximum intensity projections of rendered pseudocolored tracings of newly appearing oligodendrocytes in cortical layer I (top zone). Complete reconstructions of cell bodies (black), processes (black) and myelin sheaths are shown for control (blue, 10 cells, N = 3 mice) and cuprizone treated mice (orange, 9 cells, N = 3 mice). E-F: Histograms comparing myelin sheath number (E), total length of myelin (F) and average length of individual myelin sheaths (G) between newly formed control and remyelinating oligodendrocytes. H: Graph showing distribution of sheath lengths for individual cells 12–14 days from first appearance (Control, blue; Remyelinating, orange, with mean and SEM in gray). I: 2D montage of oligodendrocyte morphologies (cell bodies and cytoplasmic processes) illustrating process orientation. Vectors were calculated from the cell body extending to each paranode of a reconstructed oligodendrocyte at days 12–14 and *x* and *y* vector components were summed and oriented to same direction of the vector sum. J: Vector plots showing the average orientation of oligodendrocyte processes for Control (N = 10 cells from three mice) and cuprizone treated mice (N = 9 cells from three mice). Oligodendrocyte process orientations were not significantly different from uniformly radial (J, Control, –0.047 ± 1.30 (std) rad; Regenerated, 0.076 ± 1.30 (std) rad), and exhibited similar degrees of circularity. See ***Supplementary file 1*** for statistical tests and significance level for each comparison.

The online version of this article includes the following figure supplement(s) for figure 3:

**Figure supplement 1.** Dynamics of oligodendrocyte maturation in adult *Mobp-EGFP* mice.

first appearance when they reached their mature form. Although regenerated oligodendrocytes had access to much more axonal territory, they produced similar numbers of sheaths (***Figure 3E***) (Control: 53 ± 3 sheaths, 10 cells, N = 3 mice; Regenerated: 51 ± 2 sheaths, 9 cells, N = 3 mice, p=0.628, unpaired two-tailed t-test). However, regenerated cells formed more total myelin (***Figure 3F***) (Total sheath length: Control, 3.17 ± 0.16 mm; Regenerated: 3.80 ± 0.23 mm, p=0.041, unpaired two-tailed t-test) by extending slightly longer sheaths (***Figure 3G***) (average sheath length: Control, 62.6 ± 2.6 µm; Regenerated, 72.3 ± 2.2 µm, p=0.012, unpaired two-tailed t-test), despite having similar distributions of sheath lengths (***Figure 3H***).

If regenerated oligodendrocytes reach the same axonal regions from a greater distance away, their processes should be more polarized; however, 2-D maps of sheaths arising from single cells, revealed that they exhibited a similar radial morphology (***Figure 3D***). To obtain a quantitative measure of polarization, vectors were calculated from the cell body to each paranode (***Figure 3I***) and mean radial histograms for new and remyelinating cells were calculated (***Figure 3J***). The average extent of polarization for control and regenerated cells was not significantly different from uniformly radial (Control, –0.047 ± 1.30 (std) rad, p=0.400; Regenerated, 0.076 ± 1.30 (std) rad, p=0.256, Hodges-Ajne test of non-uniformity) and sheaths of new cells in control and those regenerated after cuprizone exhibited similar degrees of circularity (p>0.1, k = 462, Kuiper two-sample test). Thus, regenerated oligodendrocytes formed in an environment with greater myelination targets have morphologies remarkably similar to cells added to existing myelinated networks in naïve mice, suggesting that oligodendrocyte structure is shaped by strong cell intrinsic mechanisms.

To estimate the extent of myelin sheath recovery in the somatosensory cortex, we measured the overlap in cell territory between baseline and regenerated cells. Territories of

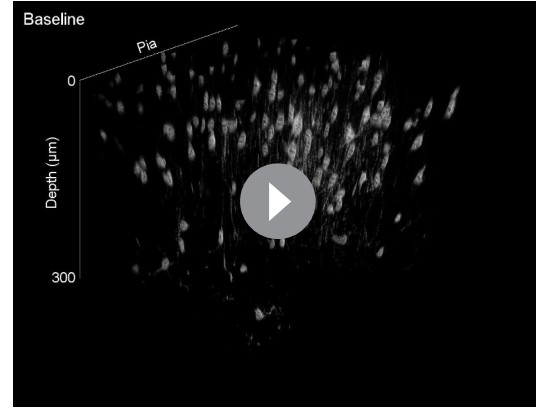

**Video 4.** New oligodendrocyte cell bodies appear in locations that are different than those of lost oligodendrocytes. Longitudinal imaging of an adult *Mobp-EGFP* mouse with a chronic cranial window fed 3 weeks of a cuprizone-supplemented diet followed through 5 weeks of recovery. All oligodendrocytes from the example region in ***Figure 4B*** (cortical volume: 425 µm (X) x 425 µm (Y) x 300 µm (Z)) are shown in three dimensions at each imaging time point, with a rotation around the Z (depth) axis. The final image is an overlay of pseudocolored cells present at baseline (magenta) or 5 weeks recovery (green), illustrating that remyelinating cells appear in distinct locations from those present at baseline.
https://elifesciences.org/articles/56621#video4

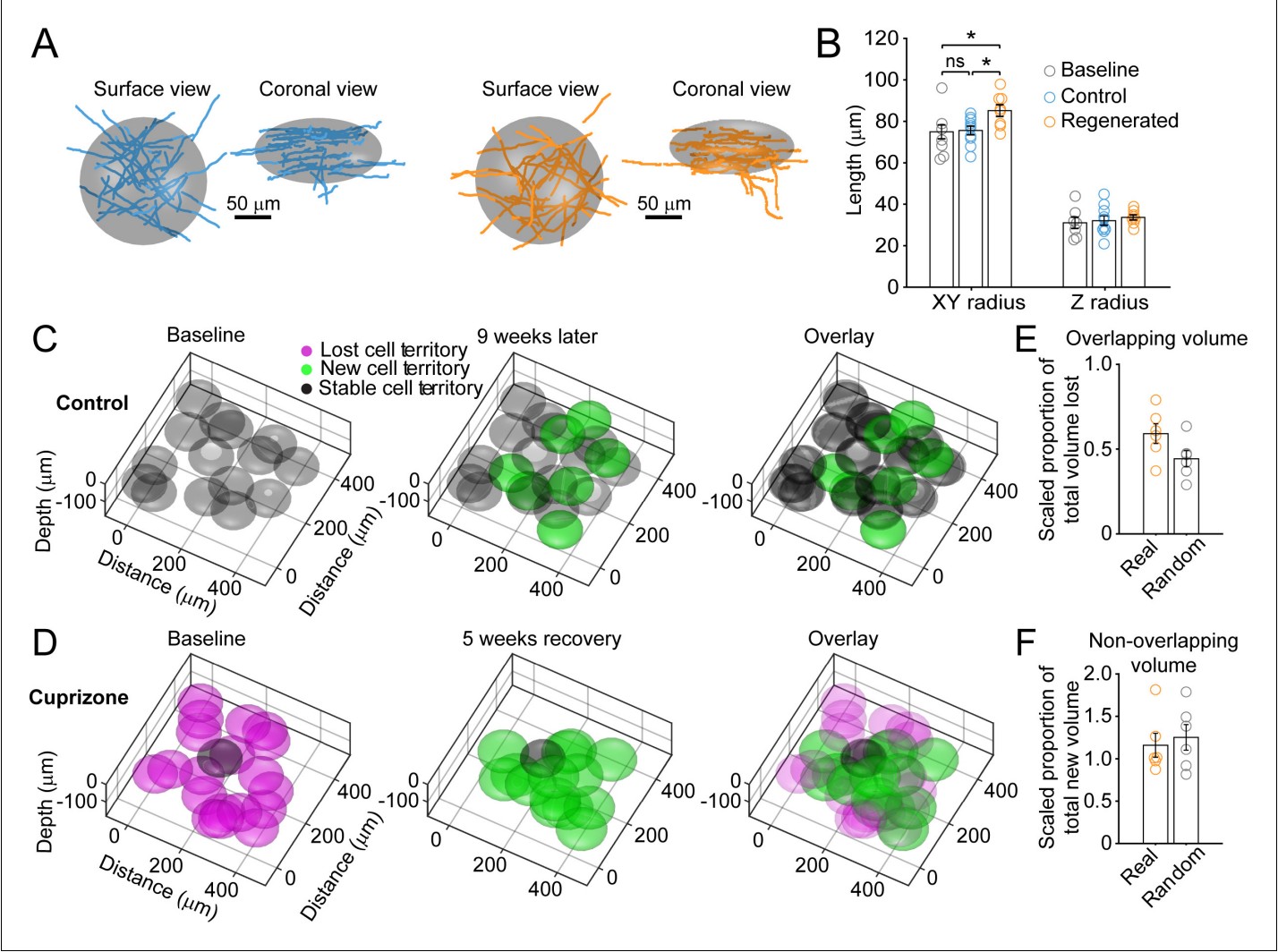

**Figure 4.** Remyelinating oligodendrocytes myelinate distinct cortical territories. (**A-B**) Schematic showing best fit ellipsoids that encompass 80% of all myelin sheaths of oligodendrocytes from Control (blue) and Remyelinating (orange) cells in the 0–100 μm zone. B: Histogram comparing ellipsoid lengths for *x-y* and *z* radius for Control (blue, N = 10 cells from three mice), Cuprizone (orange, N = 9 cells from three mice) and Baseline (gray, N = 7 cells from four mice). Circles represent individual cells. Remyelinating cells were significantly wider (*x-y* radius) than newly generated oligodendrocytes in Control. (C-D: Average best-fit ellipsoids for Control (**C**) or Remyelinating cells (**D**) calculated in (**B**) plotted for top 0–100 μm of cortex (425 μm x 425 μm x 100 μm) based on location of cell bodies. E-F: Histograms showing proportion of overlap (total territory volume) between baseline and regenerated oligodendrocytes (**E**) (scaled to account for differences in number of baseline and regenerated cells; see Materials and methods), and additional volume encompassed by regenerated oligodendrocytes relative to total baseline volume (**F**), for 0–100 μm regions in cuprizone-treated mice (orange, N = 6 mice) compared to volumes predicted if the same number of regenerated cells appeared at random (gray). Regenerated oligodendrocyte territories partially overlap with baseline volume (E; 59.1%) and encompass novel territory (F; 115%), at a similar proportion to regenerated cells placed at random. See *Supplementary file 1* for statistical tests and significance level for each comparison.

individual oligodendrocytes which existed at baseline, those generated in control, and those regenerated following cuprizone were estimated using an ellipsoid centered at the center of mass of the sheath arbor with the smallest radii in the *x-y* (oriented parallel to the pia) and *z* planes (oriented perpendicular to pia) that encompassed at least 80% of the total sheath length. These radii were averaged across all cells per condition to generate model ellipsoids (*Figure 4A*), representing the volume available to be myelinated by an individual oligodendrocyte. As expected from the slightly longer myelin sheaths produced by remyelinating cells, their average cell territory was significantly larger than control cells (*Figure 4B*) (*x-y*; Control: 75.7 ± 2.1 μm; Regenerated: 85.2 ± 2.8 μm, p=0.025, one-way ANOVA). Model ellipsoids were then centered on the cell body coordinates in

layer I from control and cuprizone-treated mice (examples shown in *Figure 4C,D*) and their degree of territory overlap determined. This analysis revealed that regenerated oligodendrocytes in cuprizone exposed mice exhibit only 59.1 ± 5.8% territory overlap (Range: 37.2–79.0%, N = 6) with oligodendrocytes that originally populated these regions of the cortex. This convergence was slightly, but not significantly, higher than predicted if the same number of regenerated cells were placed at random in the volume (44.3 ± 4.7%; p=0.078 by t-test) (*Figure 4E*), suggesting that local factors influence which progenitors differentiate. Moreover, because regenerated cell territories were larger (*Figure 4B*), they enclosed an average volume of 116 ± 14.1% of the total territory at baseline (Range: 87.4–181%) (*Figure 4F*). These data and those obtained from the displacement analysis indicate that although regenerated oligodendrocytes tend to be formed close to the sites of original cells, they unexpectedly do not completely overlap with the baseline territory; thus, regenerated oligodendrocytes are unlikely to access to the same complement of axons to myelinate, and could potentially myelinate novel axon segments.

## Regeneration of specific myelin segments

Although oligodendrocytes are formed in new locations during recovery from demyelination, they have the opportunity to regenerate specific myelin sheaths in areas where they extend processes into previously myelinated territories. To assess the extent of sheath replacement in the somatosensory cortex, we acquired high-resolution images in layer I in *Mobp-EGFP* mice, allowing assessment of the position and length of individual myelin internodes. We randomly selected 100 µm x 100 µm x 100 µm volumes in the image stacks (425 µm x 425 µm x 100 µm), and traced the full length of internodes that entered the volume at baseline and at the 5-week recovery time point in control (N = 5) and cuprizone-treated mice (N = 5) (*Figure 5A*). Internodes were classified as *stable* (supplied by the same cell at baseline and recovery), *lost* (present at baseline and absent in recovery; *Video 5*), *novel* (not present at baseline, but present in recovery; *Video 6*), or *replaced* (present at baseline, lost and then replaced by a new cell (*Video 7*). Consistent with previous findings (*Hill et al., 2018*; *Hughes et al., 2018*), in control mice almost all internodes (99.1 ± 0.5%) present at baseline remained after 5 weeks (*Figure 5A*, gray processes), demonstrating the high stability of myelin sheaths within the cortex. Generation of new oligodendrocytes within this area led to the appearance of a substantial proportion (25.8 ± 8.7%) of novel sheaths (*Figure 5A*, green processes). In mice treated with cuprizone, 84.4 ± 2.7% of myelin sheaths were destroyed in this region (*Figure 5B*), but sheath numbers were restored to baseline levels after five weeks (*Figure 5C*). However, despite this apparent recovery of overall myelin content, only 50.5 ± 2.9% of specific internodes were replaced, 32.4 ± 1.2% were lost (not replaced) and 47.6 ± 9.9% novel internodes were formed (N = 5 mice) (*Figure 5D*), indicating that regeneration leads to a dramatic change in the overall pattern of myelin within cortical circuits.

Analysis of single oligodendrocytes revealed that the degree of sheath replacement varied considerably between regions. Oligodendrocytes regenerated in regions that originally had a high density of sheaths devoted a larger proportion of their sheaths to replacement ($R^2$ = 0.4232, 13 cells, N = 4 mice) (*Figure 5E,F*). This phenomenon was most evident in cells that traversed boundaries of low and high myelin density; rather than extending all processes into the previously highly myelinated area, they retained a radial morphology (see also *Figure 3I*); processes that extended into densely myelinated areas often formed sheaths on previously myelinated axons, while processes that extended into the sparsely myelinated area typically formed novel sheaths (*Figure 5E,F*). These results suggest that remyelination may be opportunistic and based, at least initially, on chance interactions between processes and local axon segments. This finding is consistent with the highly radial morphology of premyelinating oligodendrocytes (*Trapp et al., 1997*), which would be optimized for local search of the surrounding volume rather than directed, regional targeting of subsets of axons.

Myelination patterns along axons in the cortex are highly variable, ranging from continuous to sparsely myelinated in the same region (*Hill et al., 2018*; *Hughes et al., 2018*; *Micheva et al., 2016*; *Tomassy et al., 2014*). To determine if there is a preference for replacing specific types of sheaths during recovery from demyelination, we classified sheaths present at baseline according to their neighbors: 0 neighbors within 5 µm (*isolated*), one node of Ranvier (NOR) and one hemi-node (*one neighbor*), or two NORs (*two neighbors*) (*Figure 6A–C*; *Videos 8* and *9*). In control mice, this pattern was highly stable, with similar proportions of each class of sheath present at baseline and 8 weeks later (*Figure 6D*), a pattern that we showed previously is stable through middle-age

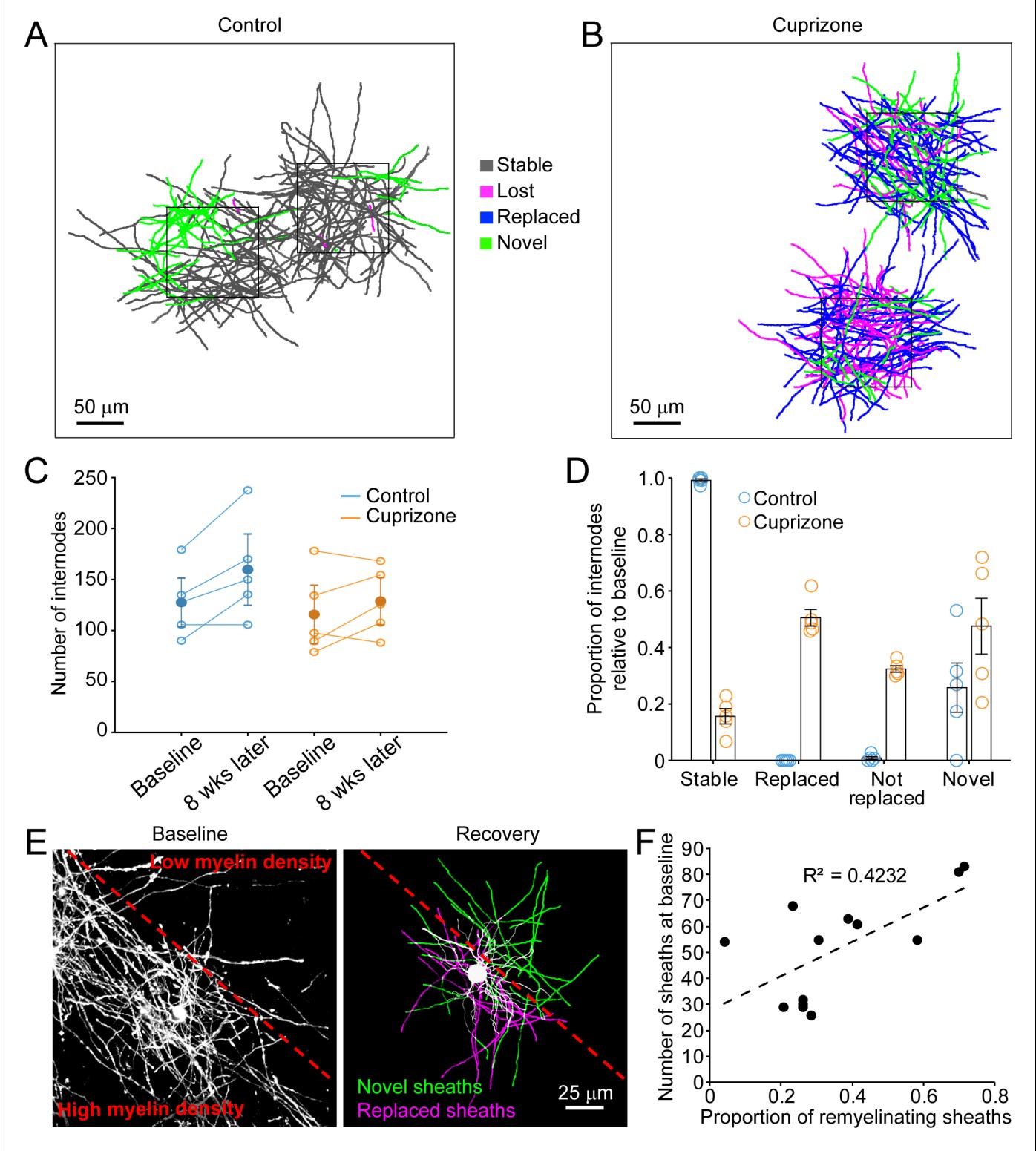

**Figure 5.** Oligodendrocyte regeneration results in a new pattern of cortical myelin. (**A–B**) 2D rendering of individual myelin sheaths that passed through a 100 μm x 100 μm x 100 μm volume within the top 0–100 μm zone. Fate of each sheath from control (**A**) and cuprizone-treated (**B**) cortex shown as stable (black), lost (magenta), replaced (blue) or novel (green) across the time series. (C) Graph of total number of traced internodes at baseline and after 8 weeks of imaging (circles represent means for individual mice, with line connecting two time-points; mean for all mice is filled circle with SEM) for control (blue, N = 5), and cuprizone-treated (orange, N = 5) mice. (D) Histogram of internode fates from cuprizone-treated (orange, N = 5) and control (blue, N = 5) mice (circles represent mean proportional values relative to baseline from individual mice). Error bars are standard error of the

*Figure 5 continued on next page*

*Figure 5 continued*

mean. (E) *Left panel*, maximum intensity projection image (226 µm x 226 µm x 60 µm) illustrating myelin sheaths at baseline. Dashed red line demarcates an area of higher (lower left) and lower myelin density (top right). *Right panel*, rendering of a regenerated oligodendrocyte that appeared at 3 days recovery, which formed myelin sheaths that either replace those lost (magenta) or are novel (green). (F) Plot showing proportion of replaced sheaths per new oligodendrocyte relative to the number of baseline myelin sheaths present within the territory (average remyelinating ellipsoid, see *Figure 4A,B*) of the new cells (13 remyelinating oligodendrocytes from N = 4 mice), correlation co-efficient $R^2$ = 0.4232.

(*Hughes et al., 2018*). Indeed, novel sheaths generated by new oligodendrocytes in control mice made roughly equal proportions of each class of sheath (*Figure 6E*), indicating that new sheaths were added to previously unmyelinated axons, as well as next to existing sheaths on myelinated axons, visible as higher values in the upper right quadrant (relative to lower left quadrant) of the myelination matrix (*Figure 6F*). We then assessed the extent of sheath replacement 5 weeks after the end of cuprizone. Similar to that observed in control, oligodendrocytes in cuprizone-treated mice had similar proportions of each class of sheath as they did at baseline (*Figure 6G*). While lost (not replaced) and novel internodes exhibited an equal proportion of isolated and neighbored sheaths, the majority of replaced internodes had at least one neighbor (79.5 ± 2.5%) (*Figure 6H*), visible in the bias toward higher values in the lower right quadrant in the myelination matrix (*Figure 6I*). These data suggest that regenerated oligodendrocytes preferentially formed sheaths on axons that were previously more heavily myelinated, suggesting that factors which promote greater myelination of specific axons are preserved after demyelination.

Analysis of individual sheath position revealed the remarkable spatial specificity with which sheaths were replaced. For both isolated sheaths and those along continuously myelinated segments, sheaths were often formed close to the same position along axons, resulting in NOR or hemi nodes at similar positions (*Figure 7A*; *Video 10*). These results suggest that there may be persistent landmarks along axons indicating the prior position of nodes after myelin is removed (*Figure 7B*). To assess whether nodal components along cortical axons remain after demyelination, we performed post-hoc immunostaining on tissue from mice exposed to cuprizone for six weeks (longer than three weeks used above, to increase the length of time that axons were devoid of myelin) with antibodies against Caspr, βIV-spectrin, and Ankyrin-G, which together label paranodal and nodal regions along myelinated axons (*Susuki et al., 2016*). High-resolution z-stacks (135 µm x 135 µm x 30 µm, 2048 × 2048 pixels) of layer V-VI were acquired in order to capture a large population of nodes (*Figure 7C,D*), as the upper layers of cuprizone-treated mice had inadequate numbers for quantification at this magnification. The full images were transformed into surfaces to allow automated quantification (see Methods) (*Figure 7E,F*). βIV-spectrin puncta were classified as 'nodes' if they were flanked by Caspr+ puncta, 'heminodes' if they were bound by only one side by Caspr, and 'isolated' if no flanking Caspr was visible (*Figure 7B*). As expected, the

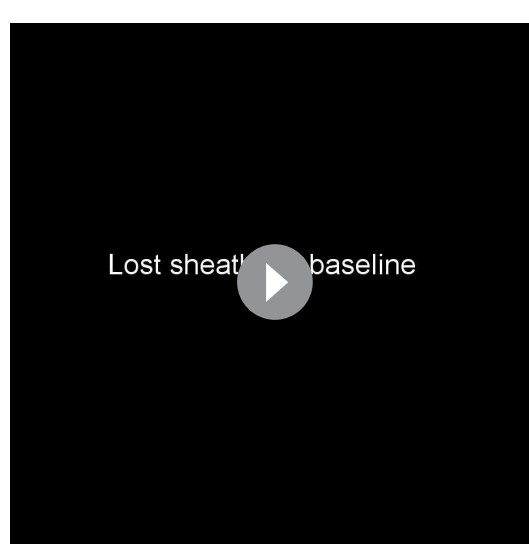

**Video 5.** Myelin sheaths are lost and not replaced over time. An example volume (100 µm x 100 µm x 100 µm) within the top 0–100 µm zone from an adult *Mobp-EGFP* mouse with a chronic cranial window who was fed 3 weeks of a cuprizone-supplemented diet followed through 5 weeks of recovery and longitudinally imaged (same region as depicted in *Figure 6C*; *Video 6 and 7*). Twenty-four individual myelin sheaths present at baseline and lost by 5 weeks of recovery are pseudocolored in magenta and overlaid with high-resolution images of EGFP-signal (white) at baseline, 3 days recovery (peak demyelination) and 5 weeks recovery time-points. For each time-point, the movie starts at the pial surface and proceeds to a depth of 100 µm in 1 µm steps. Note that magenta pseudocolored sheaths overlay EGFP+ myelin sheaths at baseline, but not at 5 weeks of recovery.
https://elifesciences.org/articles/56621#video5

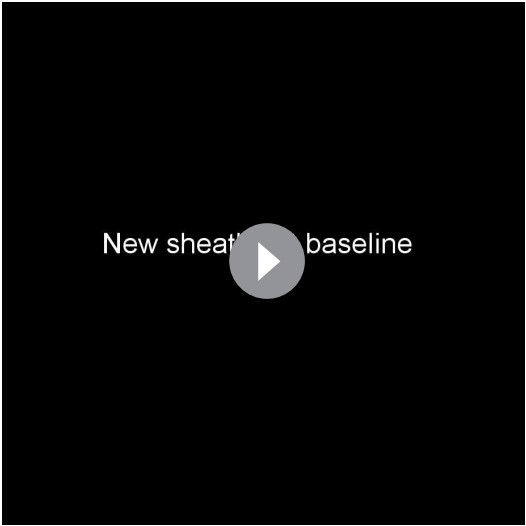

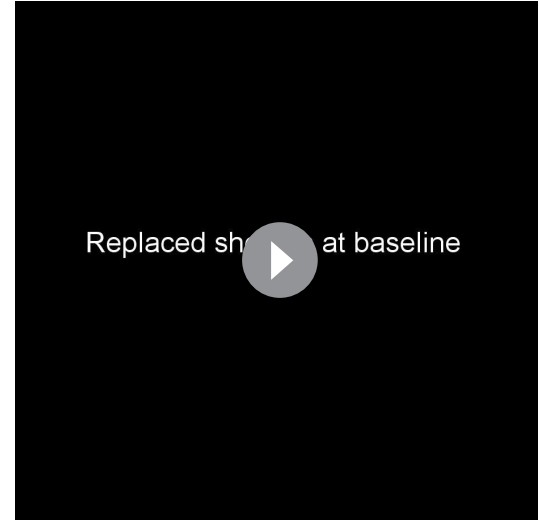

**Video 6.** Novel myelin sheaths are generated during remyelination. An example volume (100 μm x 100 μm x 100 μm) within the top 0–100 μm zone from adult *Mobp-EGFP* mouse with a chronic cranial window who was fed 3 weeks of a cuprizone-supplemented diet followed through 5 weeks of recovery and longitudinally imaged (same region as depicted in *Figure 6C*; *Video 5 and 7*). Thirty-seven individual myelin sheaths newly formed by 5 weeks of recovery, but not present at baseline, are pseudocolored in green and overlaid with high-resolution images of EGFP-signal (white) at baseline, 3 days recovery (peak demyelination) and 5 weeks recovery time-points. For each time-point, the movie starts at the pial surface and proceeds to a depth of 100 μm in 1 μm steps. Note that green pseudocolored sheaths overlay EGFP+ myelin sheaths 5 weeks of recovery, but not at baseline.
https://elifesciences.org/articles/56621#video6

**Video 7.** Individual myelin sheaths are replaced following demyelination. An example volume (100 μm x 100 μm x 100 μm) within the top 0–100 μm zone from an adult *Mobp-EGFP* mouse with a chronic cranial window who was fed 3 weeks of a cuprizone-supplemented diet followed through 5 weeks of recovery and longitudinally imaged (same region as depicted in *Figure 6C*; *Video 5 and 6*). Sixty-four individual myelin sheaths which were present at baseline, lost, and then replaced by 5 weeks of recovery are pseudocolored in blue and overlaid with high-resolution images of EGFP-signal (white) at baseline, 3 days recovery (peak demyelination) and 5 weeks recovery time-points. For each time-point, the movie starts at the pial surface and proceeds to a depth of 100 μm in 1 μm steps. Note that blue pseudocolored sheaths overlay EGFP+ myelin sheaths at baseline and at 5 weeks of recovery.
https://elifesciences.org/articles/56621#video7

distribution of nodes among these categories remained stable in control mice over two weeks (*Figure 7G*). In contrast, the characteristics of nodes changed dramatically after exposure to cuprizone, with a loss of nodes visible at 4 weeks, a time when ~ 50% of oligodendrocytes had degenerated (*Figures 1I* and *7G*). After 6 weeks in cuprizone, few Caspr+ puncta remained, and correspondingly there were few nodes or heminodes (*Figure 7F,G*). However, even at this later time point, many βIV-spectrin+ puncta were visible, with the greatest proportion of these isolated from Caspr, suggesting that βIV-spectrin remains clustered for a prolonged period. Thus, although prior studies suggest that NOR components are rapidly disassembled and redistributed after demyelination (*Coman et al., 2006*; *Craner et al., 2004*; *Dupree et al., 2004*), these results indicate that βIV-spectrin, which links the underlying actin cytoskeleton to integral membrane proteins within the node/paranode, remains clustered along axons, which may provide the means to confine sheath extension to previously myelinated regions and allow precise restoration of myelin sheath position during regeneration.

## Discussion

The organization of myelin in the cerebral cortex is remarkably diverse, with densities of myelin sheaths varying between cortical regions, between axons from different classes of neurons and even along individual axons within a given area. These patterns are established progressively over many

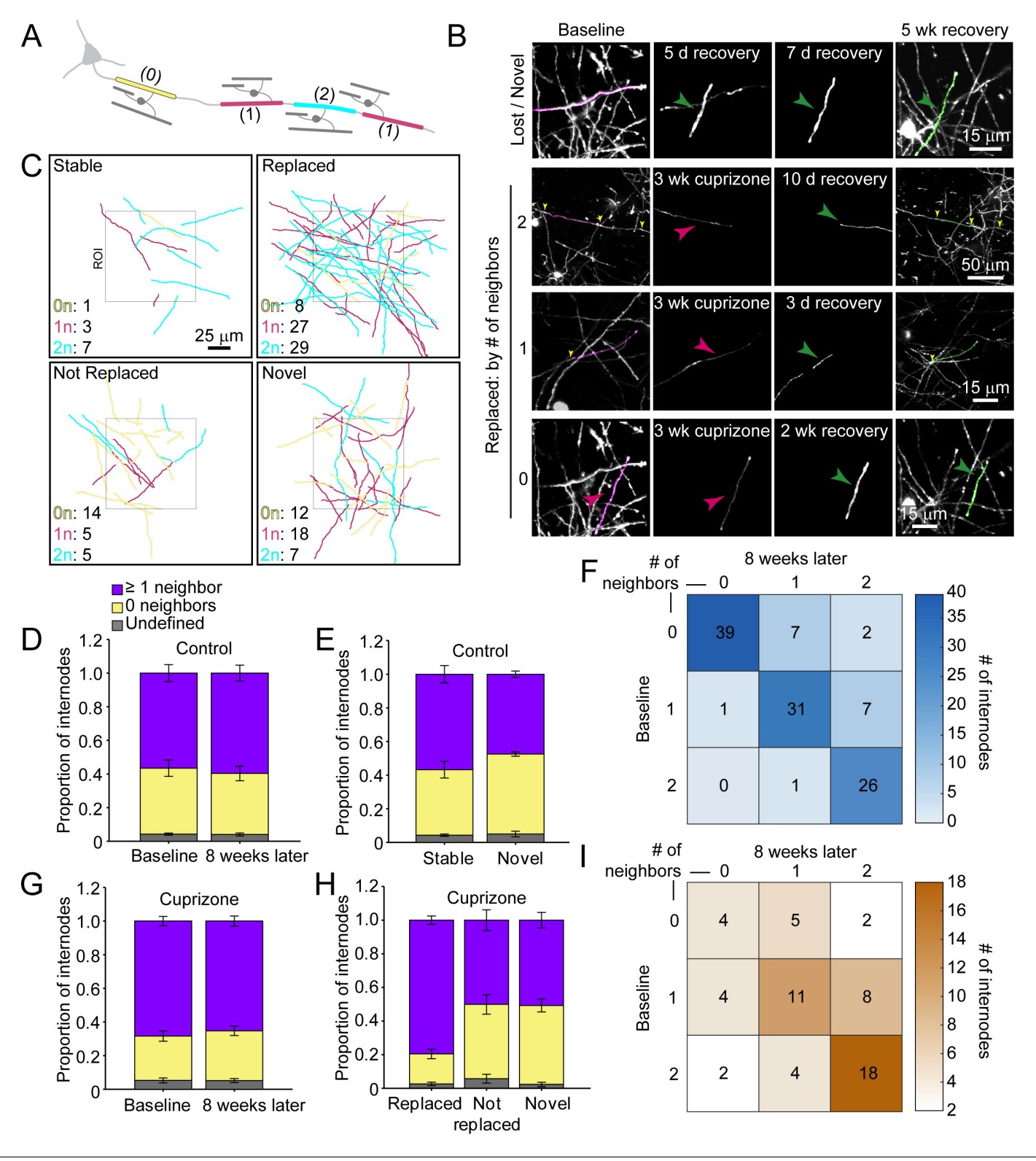

**Figure 6.** Regenerated oligodendrocytes preferentially remyelinate extensively myelinated axons. (**A**) Schematic illustrating intermittent myelination of cortical axons, designating each internode by the number of flanking myelin sheaths (0, yellow; 1, magenta; or 2, cyan). (**B**) Example maximum intensity projections of lost (magenta) and new (green) myelin sheaths from cuprizone-treated mice. Top row, example of a lost, but not replaced, sheath, and a novel sheath not present at baseline (green arrowhead; also see *Video 8*). Remaining rows are examples of lost (magenta sheaths and arrowheads) and

*Figure 6 continued on next page*

*Figure 6 continued*

replaced internodes (green sheaths and arrowheads) that had 0 (see also **Video 9**), 1, or two neighboring sheaths at baseline. Yellow arrowheads in the two neighbor example indicate nodes of Ranvier. (C) 2D rendering of traced myelin sheaths that passed through a 100 µm x 100 µm x 100 µm volume within the top 0–100 µm zone. The fate of each sheath within the volume was determined as in *Figure 5*. Sheath color illustrates whether it had 0, 1 or two neighbors as in A. (D, G) Comparison of mean proportion of internodes with at least one neighbor (lavender), isolated (yellow), or undefined (gray) within a 100 µm x 100 µm x 100 µm volume at baseline and 8 weeks later, from control (D, N = 5) and cuprizone-treated (G, N = 5) mice. Volumes from both control and cuprizone-treated conditions have the same relative proportion of isolated vs. ≥1 neighboring internode at both time-points. (E, H) Comparison of the mean proportion of internodes with 0 or ≥1 neighbor that are stable or novel (control, (E) or replaced, not replaced, or novel (cuprizone, (H). There is no significant difference in the proportion of isolated vs. ≥1 neighbor population between stable and novel sheaths in control (E), and between lost and novel sheaths in cuprizone (H), but relatively more internodes with ≥1 neighbor were replaced in cuprizone-treated cortex (H). F, I: Myelination matrix illustrating average number of internodes categorized by number of neighbors at baseline and at final imaging time-point for control (F), and cuprizone-treated mice (I). Internodes with more neighbors at baseline are more likely be replaced (largest average # of internodes in bottom right of matrix). See *Supplementary file 1* for statistical tests and significance levels for each comparison.

months, creating an extended developmental time course that results in pronounced increases in myelin through adolescence and adulthood. Although progressive, oligodendrogenesis and myelination can be enhanced by life experience and may be critical to certain forms of learning (*Gibson et al., 2014*; *Hughes et al., 2018*; *McKenzie et al., 2014*). However, once formed, oligodendrocytes and their complement of myelin sheaths are extraordinarily stable, with cell survival and sheath position varying little over months and are resistant to environmental changes such as sensory enrichment, in accordance with the high stability of myelin proteins and the persistence of oligodendrocytes in the human CNS (*Yeung et al., 2014*). Experience-dependent changes in myelin appear to occur primarily through addition of new sheaths arising from oligodendrogenesis (*Hughes et al., 2018*), although changes in myelin sheath thickness may also occur (*Dutta et al., 2018*; *Gibson et al., 2014*). The high stability of oligodendrocytes and their preferential myelination of specific neurons, such as parvalbumin interneurons in layers II/III (*Micheva et al., 2016*; *Stedehouder et al., 2017*), suggest that preserving the overall pattern of myelin is important to optimize and sustain the processing capabilities of these circuits. Consistent with this hypothesis, demyelination within the cortex in diseases such as MS is closely associated with cognitive impairment and increased morbidity. Moreover, many neurodegenerative diseases and affective disorders are associated with profound alterations in myelin (*Gouw et al., 2008*; *Ihara et al., 2010*; *Stedehouder and Kushner, 2017*), and recent studies indicate that demyelination triggers a dramatic decrease in excitatory synapses (*Araújo et al., 2017*; *Werneburg et al., 2020*), suggesting that disruptions in these sheaths may also influence both structural and functional aspects of circuits. Although preserving cortical myelination appears paramount, the highly variable patterns of myelin within cortical circuits and the sparseness of these neuron-glial associations create considerable challenges for regenerating specific myelin sheaths after injury or disease.

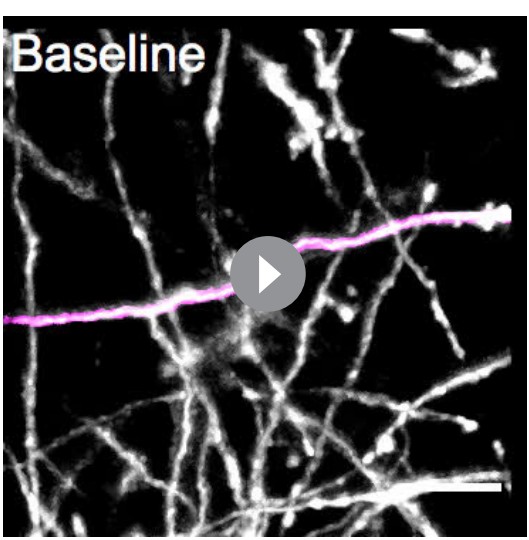

**Video 8.** Myelin sheaths are lost and novel sheaths are formed after cuprizone-treatment. Longitudinal imaging of an adult *Mobp-EGFP* mouse with a chronic cranial window fed 3 weeks of a cuprizone-supplemented diet followed through 5 weeks of recovery. A myelin sheath at baseline (traced and pseudocolored magenta, overlaid in maximum intensity projection of longitudinally-imaged region), degenerates over time (only the traced sheath from baseline is shown in subsequent time-points, and is lost by 1 week of recovery). At 5 days of recovery, a novel isolated sheath (not present at baseline, traced and pseudocolored in green in the 5 week recovery time-point overlay) appears, formed by a remyelinating oligodendrocyte not present at baseline (cell in 5-week recovery time-point overlay). Scale bar is 15 µm.
https://elifesciences.org/articles/56621#video8

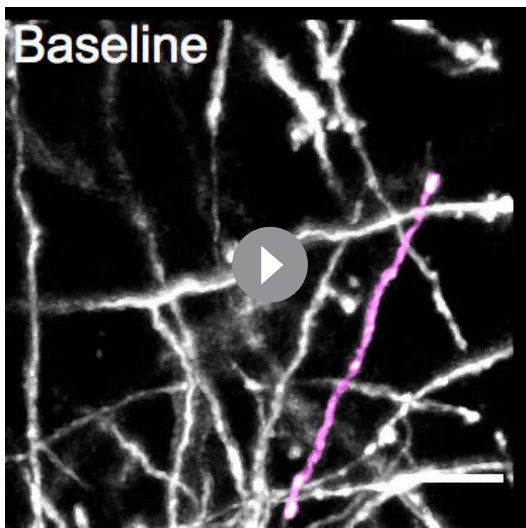

**Video 9.** Isolated myelin sheaths are replaced. Longitudinal imaging of an adult *Mobp-EGFP* mouse with a chronic cranial window fed 3 weeks of a cuprizone-supplemented diet followed through 5 weeks of recovery. An isolated myelin sheath at baseline (traced and pseudocolored magenta, overlaid in maximum intensity projection of longitudinally imaged region), degenerates over time (only the traced sheath from baseline is shown in subsequent time-points, and is lost by 3 days of recovery). At 5 days of recovery a replacement isolated sheath appears (traced and pseudocolored in the 5 week recovery time-point overlay), formed by a remyelinating oligodendrocyte not present at baseline (cell in 5-week recovery time-point overlay). Scale bar is 15 μm.

https://elifesciences.org/articles/56621#video9

Here, we used two photon in vivo imaging to examine the destruction and regeneration of oligodendrocytes and myelin sheaths in cortical circuits of adult mice. These longitudinal, high-resolution studies revealed features of the regenerative process in the cortex that were not previously known. First, oligodendrocytes were regenerated in new locations, yet had similar morphologies. Second, regenerated oligodendrocytes often formed sheaths along portions of axons that were previously unmyelinated, establishing a new pattern of myelination. Third, oligodendrocyte regeneration was not uniform across the cortex and became less efficient with depth from the cortical surface, in concert with the increasing density of myelin (prior to oligodendrocyte destruction) and enhanced gliosis. Fourth, in areas of territory overlap, regenerated oligodendrocytes were able to establish sheaths at similar positions along previously myelinated axons, indicating that positional cues persist along axons long after demyelination. Together, these findings reveal unexpected aspects of cortical remyelination, raising new questions about the mechanisms that impair oligodendrocyte regeneration in deeper cortical layers, the mechanisms that enable oligodendrocytes to identify and replace individual myelin sheaths and the long-term consequences of circuit level changes in myelination patterns.

## Regenerative potential of cortical OPCs

In regenerative processes, cell loss and cell generation are typically closely coupled, helping to ensure efficient replacement without energetically costly production of excess cells and further tissue disruption (*Biteau et al., 2011*). If this scenario were to prevail in the CNS, the generation of new oligodendrocytes should be proportional to those lost, with much higher oligodendrocyte production occurring in deeper layers. However, our results indicate that oligodendrocyte replacement was remarkably constant across layers for many weeks after a demyelinating event (*Figure 2G–I*), with regeneration in deeper layers lagging behind that predicted for one-to-one replacement. This phenomenon could be explained if local environmental factors in deeper layers suppress OPC differentiation. The larger number of oligodendrocytes that degenerate in this area may inhibit regeneration by creating more inflammation and myelin debris, factors known to suppress oligodendrogenesis. Consistent with this hypothesis, reactive gliosis was more prominent and more prolonged in deeper layers of cortex (*Figure 2—figure supplement 1A,B,E,G*). Nevertheless, even after extended recovery (>9 weeks), oligodendrocyte density remained much lower in these regions than present at baseline, without a corresponding loss of axons (*Figure 1—figure supplement 2*), although we cannot rule out the possibility that a small proportion of demyelinated axons degenerated or that pathological changes in axons impair their ability to become remyelinated (*Gudi et al., 2017*). This regional suppression of oligodendrogenesis is particularly apparent when assessing the rate of cell addition, as the initial burst in oligodendrogenesis that occurs just following cell loss is not sustained despite the remaining cell deficit (*Figure 2G–I*). We predict that it would take approximately three additional months for the oligodendrocyte population to be fully regenerated, but this is only possible if a higher rate of oligodendrogenesis (~3.5%) than observed in age-matched control brains (1.7%) is maintained. It is not clear

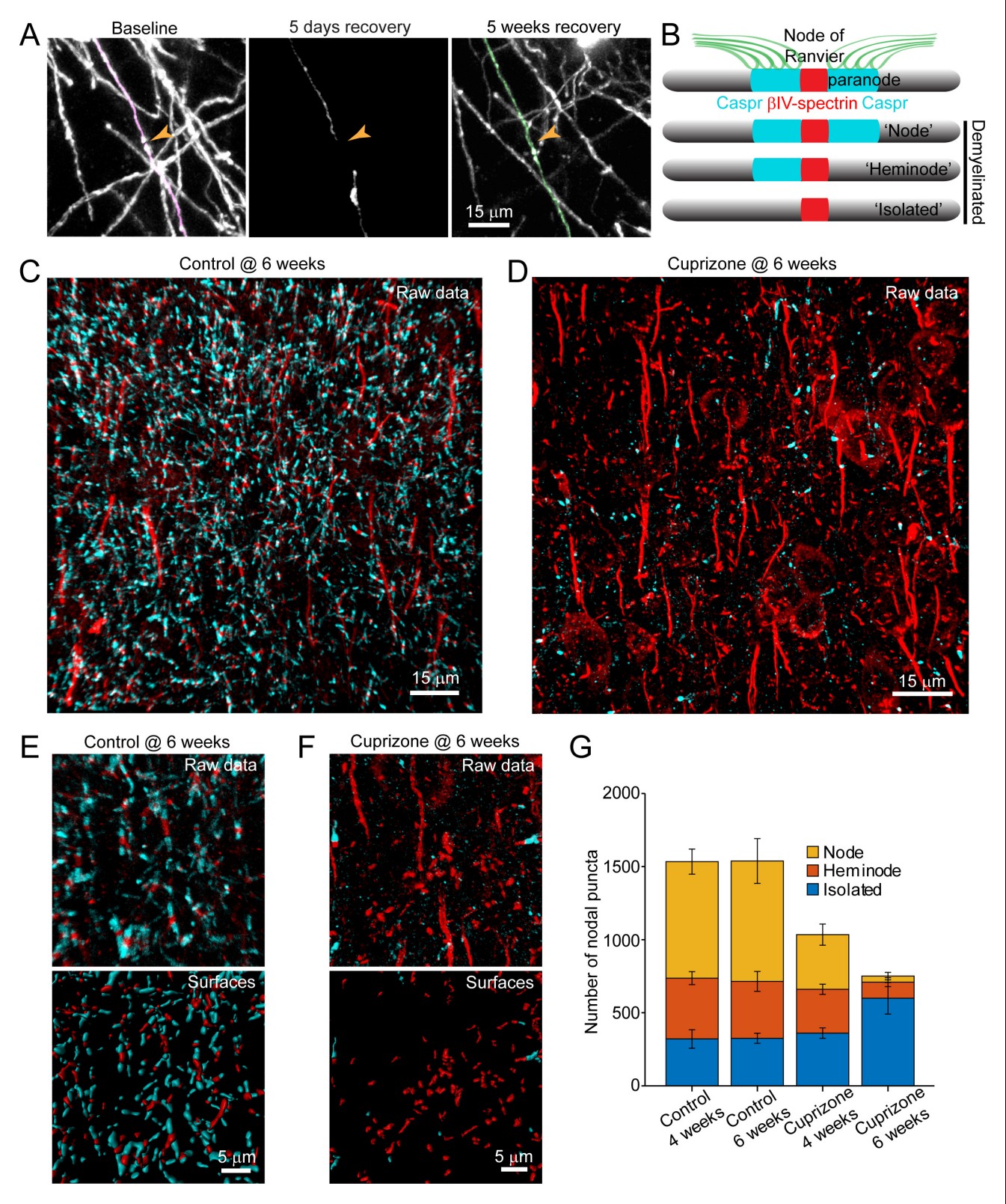

**Figure 7.** Structural components of the node of Ranvier persist after demyelination. (**A**) Images showing fate of myelin sheaths along a single axon in a cuprizone treated mouse, illustrating the loss (magenta, baseline) and regeneration of its sheaths (green, 5 weeks recovery). Pseudocolored sheaths are overlaid on maximum intensity projections from a longitudinally imaged somatosensory cortical region. Orange arrowhead denotes location of node of Ranvier at baseline and at 5 weeks of recovery (see *Video 10*). (**B**) Schematic depicting axonal regions of a myelinated axon where βIV-spectrin (node of

*Figure 7 continued on next page*

*Figure 7 continued*

Ranvier, red) and Caspr (paranode, cyan) localize. After demyelination, βIV-spectrin can be found with two flanking Caspr puncta ('Nodes'), one flanking Caspr punctum ('Heminode'), or no nearby Caspr puncta ('Isolated'). (C-D) Images βIV-spectrin and Caspr immunoreactivity in coronal sections from mice fed cuprizone-supplemented diet (D) or sham chow (C) for 6 weeks. E-F: Magnified views of βIV-spectrin and Caspr immunoreactivity (*top panels*, Raw data) from control (E) and cuprizone-treated (F) brains. Examples of post-processed (*lower panels*, Surfaces) images used to calculate nearest neighbor distances between βIV-spectrin puncta and Caspr puncta for images in E and F. Axon initial segments (AIS) were excluded from surface rendering. There was no significant difference in AIS between control (54.3 ± 2.4) and cuprizone-treated (59.3 ± 3.7) mice (p=0.354, two-tailed unpaired t-test). (G) Histogram showing total βIV-spectrin puncta categorized as either Node, Heminode or Isolated. Fewer βIV -spectrin puncta were observed after cuprizone, but isolated puncta become more numerous with increasing duration of cuprizone exposure (control @ 4 weeks, N = 3 mice; control @ 6 weeks, N = 3 mice; cuprizone @ 4 weeks, N = 4 mice; cuprizone @ 6 weeks, N = 4 mice). Bars are SEM.

whether the endogenous pool of OPCs could maintain this higher rate of differentiation (and resulting homeostatic OPC turnover) required for such a prolonged period of replacement. Indeed, it is also possible that a scarcity of key growth factors may limit the ability of these progenitors to mount an effective regenerative response. Although recent studies indicate that GLI1-expressing glial progenitors positioned within germinal zones along the lateral ventricles are mobilized in response to cuprizone-induced demyelination, forming new OPCs that migrate and differentiate into oligodendrocytes with higher probability than resident OPCs (*Samanta et al., 2015*), our results indicate that this recruitment is not sufficient in the short term to overcome existing inhibitory barriers within cortical gray matter.

The inability to recover fully from a demyelinating event raises the possibility that inflammation persists long after the initial trauma, or that OPCs in these regions are permanently altered as a result of exposure to this environment, if only for a short time (*Baxi et al., 2015*; *Kirby et al., 2019*). Like other progenitor cells, OPCs exhibit a decline in regenerative potential with age and can undergo senescence, a process that may be accelerated by exposure to inflammatory cytokines (*Kirby et al., 2019*; *Neumann et al., 2019*; *Nicaise et al., 2019*). It is also possible that there is a restricted time period during which OPCs can detect and respond to myelin loss; if there are inherent limits on OPC mobilization, as suggested by the uniform behavior of OPCs across cortical layers, then the inability to match the demand for new cells early may lead to prolonged deficits. In this regard, evidence that certain aspects of the inflammatory response strongly promote OPC differentiation (*Kotter et al., 2006*; *Miron et al., 2013*; *Ruckh et al., 2012*) raises the possibility that there is a critical time window for optimal repair. A more detailed spatial and cell-type specific profiling of inflammatory changes in the cortex after oligodendrocyte death may help clarify the role of inflammatory changes in this impaired regeneration.

An unexpected feature of oligodendrocyte regeneration in the cortex was that new cells were not formed in the same locations as the prior oligodendrocytes, even though the high density, regular spacing and dynamic motility of

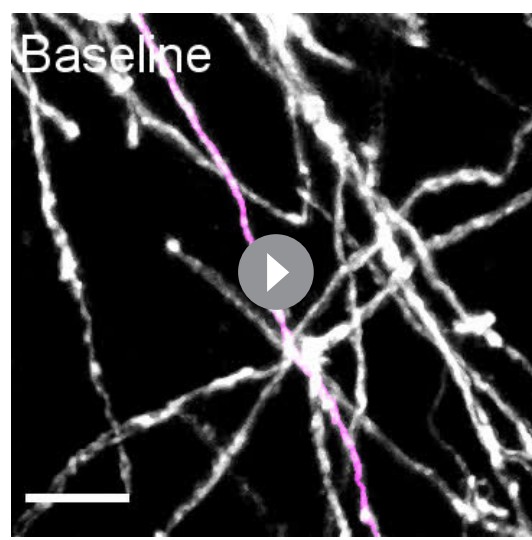

**Video 10.** Neighboring myelin sheaths with at least one neighbor are replaced and form a node of Ranvier in close proximity to one present at baseline. Longitudinal imaging of an adult *Mobp-EGFP* mouse with a chronic cranial window fed 3 weeks of a cuprizone-supplemented diet followed through 5 weeks of recovery. Two neighboring myelin sheaths (pseudocolored magenta in baseline time-point), flank an unlabeled node of Ranvier (paranodal loops of myelin accumulate cytoplasmic EGFP in *Mopb-EGFP* mice [*Hughes et al., 2018*]); these sheaths were traced over each imaging time-points, and degenerate after cuprizone-treatment. Remyelinating oligodendrocytes (one shown in 5-week recovery time-point) form replacement myelin sheaths (pseudocolored green in 5-week recovery time-point flank an unlabeled node of Ranvier in a similar position as that observed at baseline. Scale bar is 15 µm.
https://elifesciences.org/articles/56621#video10

OPCs seem ideally suited to optimize placement of new oligodendrocytes. Inhibitory factors generated as a consequence of oligodendrocyte degeneration, such as myelin debris (*Kotter et al., 2006*), axonally expressed factors (LINGO1) (*Mi et al., 2005*), extracellular matrix components (chondroitin sulfate proteoglycans) (*Lau et al., 2012*), and reactive astrocytes (*Back et al., 2005*) may create a local zone of exclusion at sites of cell death reducing the probability of OPC differentiation. It is also possible that positive, pro-myelination cues present in deeper cortical regions during development, similar to those described in the developing cerebellum (*Goebbels et al., 2017*), are no longer present in the adult cortex, rendering OPCs unable to sustain adequate remyelination.

Complete reconstruction of individual oligodendrocytes revealed that regenerated cells underwent a similar period of structural refinement over a period of ~10 days and ultimately formed a comparable number of sheaths (*Figure 3D,E*; *Figure 3—figure supplement 1A–J*). Although oligodendrocytes have the capacity to form long processes, they did not always extend their cytoplasmic processes to reach sites of original myelination, instead creating sheaths within a local territory similar to that of normal oligodendrocytes (*Figure 3D,I*), with only a 15.5% increase in total sheath length (*Figure 3F*). As regenerated oligodendrocytes are formed in an environment with an apparent surplus of receptive axons, these findings suggest that the size and shape of oligodendrocytes is profoundly limited by cell intrinsic mechanisms.

## Reparative potential of surviving oligodendrocytes

Recent studies of postmortem human tissue from MS patients have raised the intriguing possibility that remyelination may occur through the reformation of myelin sheaths by oligodendrocytes that survive autoimmune attack, rather than from de novo oligodendrogenesis (*Yeung et al., 2019*). This hypothesis is based on evidence that 'shadow plaques', which are classically considered to represent partially remyelinated axons (*Lassmann et al., 1997*), did not appear to contain many newly born oligodendrocytes, as assessed using $C^{14}$–based birth dating; the progressive decline in atmospheric $C^{14}$ levels following the cessation of atomic testing in the 1950s allow the date of last cell division to be estimated the amount of $C^{14}$ present. Although in vivo imaging studies in the adult mouse brain indicate that oligodendrocytes can be generated through direct differentiation of OPCs without cell division (*Hughes et al., 2013*), potentially confounding measures of cell age based on proliferation, these results nevertheless posit that significant new myelin may be created by existing oligodendrocytes. In our studies, the few oligodendrocytes that survived cuprizone did not contribute substantial new myelin. However, the cuprizone model does not fully recapitulate the pathology of cortical lesions observed in MS. In particular, demyelination of the upper layers of the cortex in autopsy samples from MS patients are correlated with regions of leptomeningeal inflammation, composed of B and T cells that secrete cytotoxic cytokines and create a complex inflammatory milieu (*Howell et al., 2011*; *Magliozzi et al., 2007*). Whether a cell-mediated immune response, local release of cytotoxic compounds or environmental changes substantially shift the burden of repair from OPCs to surviving oligodendrocytes in human MS remains to be explored.

## Specificity of myelin repair

The regeneration of oligodendrocytes in different locations and the strong, cell intrinsic control of cell size appear to constrain where myelin sheaths are formed. However, when a regenerated oligodendrocyte had access to a territory that was previously myelinated, it was capable of establishing sheaths along axons that were previously myelinated, indicating that the local factors which initially influenced axon selection were retained after demyelination. Here, we inferred that specific axons were remyelinated without direct axonal labeling. One of the advantages of analyzing myelin patterns in the cortex is that the trajectory of axons is highly variable, unlike the parallel axon tracks that occur in white matter. This can be appreciated from neurofilament immunostaining (*Figure 1—figure supplement 2*) and the tracings of myelin sheaths in control (*Figure 5A*). As a result, it is extremely rare for neighboring cortical axons to share a similar trajectory over ~60–70 µm, the typical length of a myelin sheath, increasing the likelihood that similarly positioned sheaths represent remyelination of the same axon.

Remarkably, myelin sheaths were often reformed at a similar position along axons, even when sheaths were isolated from neighboring sheaths and therefore could have extended over a much larger area. Although previous studies suggest that components of the NOR are redistributed after

demyelination in the PNS (*England et al., 1990*) and in MS lesions (*Coman et al., 2006*; *Craner et al., 2004*; *Dupree et al., 2004*), our studies reveal that βIV-spectrin, which forms a complex with Ankyrin-G to link voltage-gated sodium channels to the actin cytoskeleton at NORs (*Susuki et al., 2016*), remains clustered (without flanking myelin sheaths) with continued administration of cuprizone for up to six weeks (*Figure 7D,F*), suggesting that nodal guideposts remain for many weeks after demyelination. However, as these studies were performed using post-hoc immunostaining, we do not yet know if these βIV-spectrin puncta are located at previous nodal positions; future longitudinal studies using fluorescently tagged nodal components will be critical to define the stability of NORs after demyelination. The time course over which these nodal components are removed from demyelinated axons could influence the subsequent distribution of myelin, with prolonged demyelinating injuries leading to greater loss of remyelination specificity. Stabilizing the structural elements of previously myelinated axonal domains could therefore represent a potential target for remyelinating therapies. More widespread structural reorganization of demyelinated axons, resulting in changes to axonal diameter, might also alter the distribution of replaced myelin sheaths, although a recent pre-print suggests that differences in axonal diameter do not influence remyelination following oligodendrocyte ablation (*Snaidero et al., 2019*).

Because territory overlap between original and regenerated oligodendrocytes was only ~60%, considerable new axonal territory was accessed during the reparative period, leading to novel sheaths that altered the global pattern of cortical myelin. However, even in areas where the territories of regenerated and original oligodendrocytes overlapped, new myelin sheaths were often formed on portions of axons that were not previously myelinated. These findings highlight the probabilistic nature of myelination, which is influenced by many dynamic factors, such as neural activity and metabolic state that could alter target selection and stabilization of nascent sheaths (*Klingseisen et al., 2019*).

## Implications for myelin repair and cognitive function

Oligodendrocytes perform crucial roles in the CNS, enhancing the propagation of action potentials and reducing the metabolic cost to do so, providing metabolic support for axons far removed from their cell bodies and controlling excitability by influencing the distribution of voltage-gated channels and promoting clearance of extracellular potassium. Thus, the reorganization of myelin that occurs in cortical circuits during regeneration may have profound functional consequences on cognition and behavior. It will be important to determine whether these functional aspects of oligodendrocyte-neuron interactions are restored following remyelination. Our studies focused exclusively on regeneration in the somatosensory cortex of young adult mice. It is not yet known if these changes mimic regeneration in other cortical regions, although a recent pre-print indicates that remyelination in the motor cortex is similar (*Bacmeister et al., 2020* https://doi.org/10.1101/2020.01.28.923656), or if the spatial and temporal aspects of myelin replacement in the cortex vary with age. Moreover, due to limitations in resolving sheaths in deeper layers of the cortex, we do not yet know whether regeneration deficits extend to layers IV-VI. It will also be critical to define the spatial and temporal changes in myelin sheath thickness, as this varies considerably between different axons and has been shown to be a substrate for plasticity in the cortex (*Gibson et al., 2014*).

Our analysis was restricted to discrete volumes in the cortical mantle. As we are not yet able to monitor myelination patterns along the entire length of individual axons, it is possible that the position of sheaths may change after regeneration, but that the overall myelin content along a given axon is conserved. Alternatively, from a functional standpoint, it may not be necessary to reform the precise pattern of myelination, as long as the relative amount of myelin along different classes of neurons is preserved. At present, our ability to predict the consequences of these changing myelin patterns and the spatial differences in oligodendrocyte regeneration are limited by our knowledge about the functions of myelin in cortical gray matter. As recent studies suggest that even subtle changes in oligodendrogenesis can alter behavioral performance (*McKenzie et al., 2014*; *Xiao et al., 2016*), the impact of these changes may be profound. The ability to monitor the regeneration of myelin sheaths with high spatial and temporal resolution in vivo within defined circuits provides new opportunities to evaluate the effectiveness of potential therapeutic interventions (*Deshmukh et al., 2013*; *Early et al., 2018*; *Mei et al., 2014*; *Rankin et al., 2019*) and a platform to explore the functional consequences of myelin reorganization.

# Materials and methods

## Key resources table

| Reagent type (species) or resource | Designation | Source or reference | Identifiers | Additional information |
|---|---|---|---|---|
| Stain, strain background (*M. musculus*) | STOCK Tg(Mobp-EGFP) IN1Gsat/Mmucd | MMRC | RRID:MMRRC_030483- UCD | *Hughes et al., 2018* |
| Stain, strain background (*M. musculus*) | C57BL/6NCrl | Charles River | RRID:IMSR_CRL:27 | |
| Antibody | Rabbit anti-Aspartoacylase (ASPA) | Genetex | RRID:AB_2036283 | 1:1500 |
| Antibody | Goat anti-GFP | Cell signalling | RRID:AB_10705523 | 1:500 |
| Antibody | Chicken anti-GFP | Aves lab | RRID:AB_2307313 | 1:1500 |
| Antibody | Rabbit anti-GFP | Richard Huganir lab | Gift of R. Huganir | 1:1000 |
| Antibody | Mouse anti-MBP | Sternberger | RRID:AB_2564741 | 1:2000 |
| Antibody | Chicken anti-MBP | Aves | RRID:AB_2313550 | 1:500 |
| Antibody | Guinea pig anti-NG2 | Generated in D.E. Bergles lab against entire NG2 protein *Kang et al., 2013* | | 1:10,000 |
| Antibody | Rabbit anti- $\beta$IV spectrin | Generated by M. Rasband Lab | Gift from M. Rasband | 1:300 |
| Antibody | Chicken anti- $\beta$IV spectrin | Generated by M. Rasband Lab | Gift from M. Rasband | 1:100 |
| Antibody | Rabbit anti-Ankrin G | Generated by M. Rasband Lab | Gift from M. Rasband | 1:200 |
| Antibody | Guinea pig anti-Caspr | Generated by M. Bhat Lab | Gift from M. Bhat | 1:1500 |
| Antibody | Rabbit anti-Neurofilament-L | Cell Signaling | RRID:AB_823575 | 1:1000 |
| Antibody | Donkey anti-Rabbit conjugated to Alexa 488, Cy3 or Cy5 | Jackson Immuno | RRID:AB_2340619 RRID:AB_2313568 RRID:AB_2340625 | 1:2000 |
| Antibody | Donkey anti- Mouse conjugated to Alexa 488, Cy3 or Cy5 | Jackson Immuno | RRID:AB_2340849 RRID:AB_2340817 RRID:AB_2340820 | 1:2000 |
| Antibody | Donkey anti- Guinea pig conjugated to Alexa 488, Cy3 or Cy5 | Jackson Immuno | RRID:AB_2340454 RRID:AB_2340461 RRID:AB_2340477 | 1:2000 |
| Antibody | Donkey anti- Goat conjugated to Alexa 488 or Cy3 | Jackson Immuno | RRID:AB_2340430 RRID:AB_2340413 | 1:2000 |
| Antibody | Donkey anti- Chicken conjugated to Alexa 488 or Cy5 | Jackson Immuno | RRID:AB_2340376 RRID:AB_2340347 | 1:2000 |
| Chemical compound, drug | bis(cyclohexanone) oxaldihydrazone (Cuprizone) | Sigma-Aldrich | Catalog # C9012 | |
| Software, algorithm | ZEN Blue/Black | Zeiss | RRID:SCR_013672 | |
| Software, algorithm | Fiji | http://fiji.sc | RRID:SCR_002285 | |
| Software, algorithm | ImageJ | https://imagej.nih.gov/ij/ | RRID:SCR_003070 | |
| Software, algorithm | Adobe Illustrator CS6 | Adobe | RRID:SCR_014198 | |
| Software, algorithm | MATLAB | Mathworks | RRID:SCR_001622 | |

*Continued on next page*

*Continued*

| Reagent type (species) or resource | Designation | Source or reference | Identifiers | Additional information |
|---|---|---|---|---|
| Software, algorithm | SyGlass | IstoVisio | RRID:SCR_017961 | |
| Software, algorithm | Imaris | Bitplane | RRID:SCR_007370 | |

## Animal care and use

Female and male adult mice were used for experiments and randomly assigned to experimental groups. All mice were healthy and did not display any overt behavioral phenotypes, and no animals were excluded from the analysis. Generation and genotyping of BAC transgenic lines from *Mobp-EGFP* (Gensat) have been previously described (*Hughes et al., 2018*). Mice were maintained on a 12 hr light/dark cycle, housed in groups no larger than 5, and food and water were provided ad libitum (except during cuprizone-administration, see below). All animal experiments were performed in strict accordance with protocols approved by the Animal Care and Use Committee at Johns Hopkins University.

## Cranial windows

Cranial windows were prepared as previously described (*Holtmaat et al., 2012*; *Hughes et al., 2018*). Briefly, mice 7 to 10 weeks old were anesthetized with isoflurane (induction, 5%; maintenance, 1.5–2%, mixed with 0.5 L/min $O_2$), and their body temperature was maintained at 37° C with a thermostat-controlled heating plate. The skin over the right hemisphere was removed and the skull cleaned. A 2 × 2 or 3 × 3 mm region of skull over somatosensory cortex ($-1.5$ mm posterior and 3.5 mm lateral from bregma) was removed using a high-speed dental drill. A piece of cover glass (VWR, No. 1) was placed in the craniotomy and sealed with VetBond (3M), then dental cement (C and B Metabond) and a custom metal plate with a central hole was attached to the skull for head stabilization.

## In vivo two photon microscopy

In vivo imaging sessions began 2 to 3 weeks after cranial window procedure (Baseline). After the baseline imaging session, mice were randomly assigned to cuprizone or control conditions. During imaging sessions, mice were anesthetized with isoflurane and immobilized by attaching the head plate to a custom stage. Images were collected using a Zeiss LSM 710 microscope equipped with a GaAsP detector using a mode-locked Ti:sapphire laser (Coherent Ultra) tuned to 920 nm. The average power at the sample during imaging was <30 mW. Vascular landmarks were used to identify the same cortical area over longitudinal imaging sessions. Image stacks were 425 µm x 425 µm x 110 µm (2048 × 2048 pixels, corresponding to cortical layer I, Zeiss 20x objective), 425 µm x 425 µm x 550 µm (1024 × 1024 pixels) or 850 µm x 850 µm x 550 µm (1024 × 1024 pixels; corresponding to layers I – IV), relative the cortical surface. Mice were imaged every 1 to 7 days, for up to 15 weeks.

## Cuprizone administration

At 9 to 11 weeks of age, male and female *Mobp-EGFP* or C57BL/6 mice were fed a diet of milled, irradiated 18% protein rodent diet (Teklad Global) alone (control) or supplemented with 0.2% w/w bis(cyclohexanone) oxaldihydrazone (Cuprizone, Sigma-Aldrich) in custom gravity-fed food dispensers for 3 to 6 weeks. Both control and experimental condition mice were returned to regular pellet diet during the recovery period (*Baxi et al., 2017*).

## Immunohistochemistry

Mice were deeply anesthetized with sodium pentobarbital (100 mg/kg b.w.) and perfused transcardially with 4% paraformaldehyde (PFA in 0.1 M phosphate buffer, pH 7.4). Brains were then postfixed in 4% PFA for 12 to 18 hr, depending on antibody sensitivity to fixation, before being transferred to a 30% sucrose solution (in PBS, pH 7.4). For horizontal sections, cortices were flat-mounted between glass slides and postfixed in 4% PFA for 6 to 12 hr at 4°C, transferred to 30% sucrose solution (in PBS, pH 7.4). Tissue was stored at 4°C for more than 48 hr before sectioning. Brains were

extracted, frozen in TissueTek, sectioned (−1.5 mm posterior and 3.5 mm lateral from bregma) at 30 to 50 µm thickness on a cryostat (Thermo Scientific Microm HM 550) at −20°C. Immunohistochemistry was performed on free-floating sections. Sections were preincubated in blocking solution (5% normal donkey serum, 0.3% Triton X-100 in PBS, pH 7.4) for 1 or 2 hr at room temperature, then incubated for 24 to 48 hr at 4°C or room temperature in primary antibody (listed in Key Resources Table). Secondary antibody (see Key Resources Table) incubation was performed at room temperature for 2 to 4 hr. Sections were mounted on slides with Aqua Polymount (Polysciences). Images were acquired using either an epifluorescence microscope (Zeiss Axio-imager M1) with Axiovision software (Zeiss) or a confocal laser-scanning microscope (Zeiss LSM 510 Meta; Zeiss LSM 710; Zeiss LSM 880). For glial cell counts, individual images of coronal sections were quantified by a blinded observer for number of NG2+, GFAP+ and EGFP+ cells within a 425 µm x 500 µm region, and divided into 425 µm x 100 µm zones from the pial surface (*Figure 2—figure supplement 1*). Immunostaining for nodal components was performed as above, except mice were transcardially perfused with PBS only and post-fixed in 4% PFA for 50 min. Immunostaining for neurofilament-L was performed as above, except frozen sections were acquired at 7 µm thickness directly onto slides. Immunohistochemistry solutions were applied onto slides at standard concentrations and times as free-floating sections described above. A combination of *Mobp-EGFP* and C57BL/6 mice were used for histological analyses.

## Image processing and analysis

Image stacks and time series were analyzed using FIJI/ImageJ. For presentation in figures, image brightness and contrast levels were adjusted for clarity. Myelin sheath images were additionally denoised with a 3-D median filter (radius 0.5 to 1.5 pixels). Longitudinal image stacks were registered using FIJI plugin 'Correct 3D Drift' (*Parslow et al., 2014*) and then randomized for analysis by a blinded observer. Neurofilament-L images were acquired by confocal microscopy using a 20x objective and consistent pinhole (0.84 Airy units) across sections. Laser power and detector gain were adjusted by a blinded experimenter on a slice-to-slice basis to optimize brightness and contrast. Single optical slices were imported into ImageJ and processed with plugin 'Background Subtraction' with a 20-pixel rolling ball. Images were subsequently binarized using Bernsen's method with the 'Auto Local Threshold' plugin using a 20-pixel radius and a gray value contrast threshold of 15.

## Cell tracking

Individual oligodendrocytes were followed in four dimensions using custom FIJI scripts (*Hughes et al., 2018*) or with SyGlass (IstoVisio) virtual reality software by defining individual EGFP+ cell bodies at each time point, recording *xyz* coordinates, and defining cellular behavior (new, lost, or stable cells). Oligodendrocytes that were characterized as 'lost' initially lost EGFP signal in processes and myelin sheaths, before complete loss of signal from the cell body position. A 'new' oligodendrocyte appeared with novel processes and internodes absent in baseline images. Dynamics of cell body positions were analyzed with custom MATLAB scripts (https://github.com/clcall/Orthmann-Murphy_Call_etal_2020_Elife; copy archived at https://github.com/elifesciences-publications/Orthmann-Murphy_Call_etal_2020_Elife; *Call, 2020*), and cross-time point comparisons of 3-D coordinates were corrected by adding the average vector of movement of all cells between those timepoints (to account for imperfect image registration and expansion/contraction of brain volume over time). For quantification between different 100 µm depths, cells were binned between planes horizontal to the plane of the pia, and included cells were found by Delaunay triangulation.

## Myelin sheath analysis

Registered longitudinal in vivo Z-stacks collected from *Mobp-EGFP* mice were acquired using two-photon microscopy. Similar to that described previously (*Hughes et al., 2018*), all myelin sheaths within a volume of 100 µm x 100 µm x 100 µm from the pial surface were traced in FIJI using Simple Neurite Tracer (*Longair et al., 2011*) at the baseline or final recovery time-point. Then, using registered time-series images from baseline to final recovery time-point, each myelin sheath was categorized as having 0, 1 or 2 myelin sheath neighbors (*Figure 6*), whether it was stable (connected via cytoplasmic process to same cell at baseline and at 5 weeks recovery), lost (present at baseline, but not at 5 weeks recovery), replaced (≥50% of the original sheath length was replaced by a sheath

connected via cytoplasmic process to a regenerated oligodendrocyte), or novel (a sheath not present at baseline that was connected to a regenerated oligodendrocyte). If it was a stable or replaced myelin sheath, we determined whether the baseline myelin sheath had the same or different number of neighboring myelin sheaths. Myelin sheaths within the field that could not be definitively categorized were classified as 'undefined'. Myelin paranodes were identified by increased EGFP fluorescence intensity (*Hughes et al., 2018*). Nodes of Ranvier were confirmed by plotting an intensity profile across the putative node; if the profile consisted of two local maxima separated by a minimum less than that of the internode, and the length of the gap between EGFP+ processes was <5 µm, the structure was considered a node. For each field, myelin sheaths were traced by one investigator and independently assessed by a second investigator.

## Analysis of temporal and spatial dynamics of individual oligodendrocytes

Registered longitudinal in vivo Z-stacks collected from *Mobp-EGFP* mice were acquired using two-photon microscopy every 1–3 days to follow the dynamics of newly formed mature oligodendrocytes within cortical layer I at high resolution (200–400 µm *x-y*, 100–120 µm *z*; 2048 × 2048 pixels). Using images from day of appearance, all processes originating from the cell body, branch points, and individual myelin sheaths were traced in FIJI using Simple Neurite Tracer (*Longair et al., 2011*). Traced segments were put through a smoothing function prior to length calculations to reduce artifacts of jagged traces. The fate of each process and myelin sheath (stable, lost) and changes in length (stable, growth, retraction) were determined for up to 14 days per cell.

## Single oligodendrocyte territory analysis

3-D coordinates of traces from all sheaths of a single oligodendrocyte were averaged to find the center of mass (to reduce artifactual territory volume that would be above the pia resulting from centering at the cell body). From this center, ellipsoid volumes were calculated from *x-y* and *z* radii, working backwards from the distance of the furthest sheath voxel in 1-µm increments in each dimension. The ellipsoid dimensions for each traced cell was determined by the combination of *x-y* and *z* radii that produced the smallest volume containing $\geq$ 80% of the sheath voxels for that cell. Voxels within the ellipsoid were calculated by $\frac{x_i}{r_x} + \frac{y_i}{r_y} + \frac{z_i}{r_z} \leq 1$ where $(x_i, y_i, z_i)$ is a voxel belonging to a sheath, and $r_x$, $r_y$, and $r_z$ are the radii being tested. Because all traced cell morphologies were not found to be significantly different from radial, the x and y radii were held equivalent. The average *x-y* and *z* radii across all traced baseline, control, or regenerated cells were used as standard dimensions for an oligodendrocyte 'territory' in subsequent calculations.

Total territory volume for a single time-point (either baseline or 5 weeks recovery) was calculated in the following manner. In a 3-D matrix with dimensions, 425 µm x 425 µm x 33 µm corresponding to the number of voxels in the top zone of the imaged volume, all voxels contained within the territories of each cell were represented as 1's, and all voxels outside of cell territories were represented as 0's. Voxels within territories of more than one cell had values equal to the number of territories they were within. Total baseline volume 'replaced' by regenerated territories was calculated as: ((5-week recovery matrix – baseline matrix) $\leq$ 0) + baseline matrix. The total proportion of baseline volume overlapped by regenerated territories was then calculated as: 'replaced volume'/baseline volume. Because each imaged region had varying numbers of oligodendrocytes at baseline and regenerated oligodendrocytes in recovery, final territory overlap proportions were scaled in the following manner: total overlap × (number of cells @ baseline ÷ number of cells @ 5 weeks recovery). Comparisons against randomly-placed territories used *x*, *y*, and *z* coordinates each generated from a random uniform distribution of values within the imaged volume. The number of random 3-D coordinates was equivalent to the number of regenerated oligodendrocytes per volume for statistical comparisons.

## Analysis of nodal components

Following immunostaining, images were acquired at 63x on a Zeiss 880 microscope at high resolution (135 × 135×40 µm, 2048 × 2048 pixels). Images were processed in FIJI (background subtraction, rolling ball 60 pixels; 3-D median filter, 1.2 pixels XY 0.5 pixels Z). Images were acquired from sections immunostained for Caspr and βIV-spectrin, or both βIV-spectrin and Ankyrin-G. For images

with both nodal components labeled, we found complete overlap of signal. In these cases, images were processed with 'Image Calculator' in FIJI to reduce noise between channels, subtracting Ankyrin-G from βIV-spectrin signal. Images were imported into IMARIS and 3-D positions of all nodal signal and paranodal puncta (Caspr) were resolved with low- and high-pass filters and 'Surface' and 'Spot' functions. Axon initial segments were excluded using a size cutoff of $\leq$6 µm. Custom MATLAB scripts were used to detect proximity (nearest neighbor Euclidian distance, threshold 3.5 µm radius) of nodal puncta to Caspr puncta to determine proportions of nodes of Ranvier (2 Caspr puncta within 3.5 µm), heminodes (1 Caspr punctum within 3.5 µm), or isolated (no nearby Caspr puncta). Coordinates from one channel were rotated 90 degrees in the *x-y* plane before running the proximity analysis again to confirm observed proportions were not due to chance.

## Statistical analysis

No statistical tests were used to predetermine sample sizes, but our sample sizes are similar to those reported in previous publications (*Hughes et al., 2018*). Statistical analyses were performed with MATLAB (Mathworks) or Excel (Microsoft). Significance was typically determined using N-way ANOVA test with Bonferroni correction for multiple comparisons.The statistical tests used to measure significance and the corresponding significance level (typically p value) are listed in *Supplementary file 1* and source data included in Source data file 1. N represents the number of animals used in each experiment, unless otherwise noted. Data are reported as mean ± SEM and $p<0.05$ was considered statistically significant. Level of significance is marked on figures as follows: *=$p < 0.05$; **=$p < 0.01$; ***=$p < 0.001$; ****=$p < 0.0001$.

## Data availability

All source data reported in the text and figures are contained in *Source data 1*.

# Acknowledgements

We thank Dr. M Pucak and N Ye for technical assistance, T Shelly for machining expertise, M Bhat for gift of the anti-Caspr antibody and members of the Bergles laboratory for discussions. J Orthmann-Murphy was supported by grants from the National Multiple Sclerosis Society and the Hilton Foundation. C Call and G Molina-Castro were supported by National Science Foundation Graduate Research Fellowships (DGE-1746891). Funding was provided by grants from the NIH (NS051509, NS050274, NS080153), a Collaborative Research Center Grant from the National Multiple Sclerosis Society, and the Dr. Miriam and Sheldon G Adelson Medical Research Foundation to D Bergles.

# Additional information

## Competing interests

Dwight E Bergles: Reviewing editor, *eLife*. Peter A Calabresi: PI on grants to JHU from Biogen and Annexon and has received consulting fees for serving on scientific advisory boards for Biogen and Disarm Therapeutics. 1147. The other authors declare that no competing interests exist.

## Funding

| Funder | Grant reference number | Author |
|---|---|---|
| Conrad N. Hilton Foundation | | Jennifer Orthmann-Murphy |
| National Multiple Sclerosis Society | | Jennifer Orthmann-Murphy Peter A Calabresi |
| National Science Foundation | Graduate Research Fellowship | Cody L Call Gian C Molina-Castro |
| National Institutes of Health | NS051509 | Dwight E Bergles |
| National Institutes of Health | NS050274 | Dwight E Bergles |
| National Institutes of Health | NS080153 | Dwight E Bergles |

Dr. Miriam and Sheldon G Adelson Medical Research Foundation

Dwight E Bergles

The funders had no role in study design, data collection and interpretation, or the decision to submit the work for publication.

## Author contributions

Jennifer Orthmann-Murphy, Cody L Call, Conceptualization, Data curation, Formal analysis, Supervision, Funding acquisition, Validation, Investigation, Visualization, Methodology, Writing - original draft, Writing - review and editing; Gian C Molina-Castro, Data curation, Formal analysis, Funding acquisition, Validation, Investigation, Visualization, Methodology, Writing - review and editing; Yu Chen Hsieh, Formal analysis; Matthew N Rasband, Resources, Methodology, Writing - review and editing; Peter A Calabresi, Conceptualization, Methodology, Writing - review and editing; Dwight E Bergles, Conceptualization, Resources, Supervision, Funding acquisition, Investigation, Methodology, Writing - original draft, Project administration, Writing - review and editing

## Author ORCIDs

Cody L Call https://orcid.org/0000-0003-2254-4298
Gian C Molina-Castro http://orcid.org/0000-0002-0700-4042
Matthew N Rasband https://orcid.org/0000-0001-8184-2477
Dwight E Bergles https://orcid.org/0000-0002-7133-7378

## Ethics

Animal experimentation: This study was performed in accordance with the recommendations provided in the Guide for the Care and Use of Laboratory Animals of the National Institutes of Health. All experiments and procedures were approved by the Johns Hopkins Institutional Care and Use Committee (protocols: MO17M268, MO17M338). All surgery was performed under isoflurane anesthesia and every effort was made to minimize suffering.

## Decision letter and Author response

Decision letter https://doi.org/10.7554/eLife.56621.sa1
Author response https://doi.org/10.7554/eLife.56621.sa2

## Additional files

### Supplementary files

• Source data 1. Source data for figures.
• Supplementary file 1. Summary of statistical tests and significance level for comparisons by comparison for each referenced figure panel.
• Transparent reporting form

### Data availability

All data generated or analyzed in this study are included in the manuscript. Source code for analysis and figure generation are located at: https//github.com/clcall/Orthmann-Murphy_Call_etal_2020_Elife (copy archived at https://github.com/elifesciences-publications/Orthmann-Murphy_Call_etal_2020_Elife).

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
