## [Decision Letter]

**Acceptance summary:**

The authors used time-series two-photon in vivo imaging to examine how oligodendrocytes myelinate axons in the gray matter after cuprizone-induced demyelination. This topic has been difficult to address with more conventional approaches, yet is very important for understanding remyelination in the CNS. Their analysis revealed that remyelination was more efficient in the upper layers of the cortex, and that some oligodendrocytes appear in new locations, where they form sheaths at positions of the axons that were previously unmyelinated, thereby forming new patterns of myelination. The paper is full of new insights into cortical remyelination that raise many important questions for future research.

**Decision letter after peer review:**

Thank you for submitting your article "Remyelination alters the pattern of myelin in the cerebral cortex" for consideration by *eLife*. Your article has been reviewed by three peer reviewers, and the evaluation has been overseen by a Reviewing Editor and Gary Westbrook as the Senior Editor. The following individuals involved in review of your submission have agreed to reveal their identity: Robert H Miller (Reviewer #1); Mikael Simons (Reviewer #2); Ben Emery (Reviewer #3).

The reviewers have discussed the reviews with one another and agree that only modest revisions are necessary. Congratulations on such positive reviews. As the comments are straightforward, the original reviews are included below. Please include a point-by-point response to these points with your revision.

Reviewer #1:

This is a very interesting and well-done manuscript that provides detailed insights into the regenerative capacity of the oligodendrocyte lineage in response to gray matter demyelination. Using the cuprizone model the authors clearly demonstrate that while myelin sheaths are highly stable in the undamaged adult, following demyelination/remyelination the pattern, axonal selection and number of myelin sheaths and oligodendrocytes are altered. There are a number of interesting aspects to the work including the description of local differences in the extent of repair and the notion that there is a residual "axonal mark" of nodes of Ranvier. By its nature this is a descriptive manuscript, however this in no way detracts from its importance as a foundation for further mechanistic studies. Overall the analyses are detailed, rigorous and appropriately applied. I have no major issues with the current manuscript.

Reviewer #2:

This is an outstanding paper on the regeneration of myelinating oligodendrocytes in the cortex of mice. The authors use longitudinal two photon in vivo imaging to examine how oligodendrocytes myelinate axons in the cortex after cuprizone induced demyelination. This analysis led to a number of new exciting discoveries: Remyelination was more efficient in the upper layers of the cortex. Some oligodendrocytes appear in new locations, where they form sheaths at positions of the axons that were previously unmyelinated, thereby forming new pattern of myelination. They also found that oligodendrocytes preferentially formed sheaths on axons that were previously myelinated. Analysis of sheath position revealed spatial specificity, possibly by βIV-spectrin marked landmarks along the axon. Thus, the paper is full of new important insights into cortical remyelination that raise many important questions for future research. The paper is very well-written and technically convincing. I recommend publication with only few suggestions for the revision:

1) The Discussion is well written, but the possibility that axonal damage may have occurred during cuprizone treatment is not mentioned. This may, at least partly, have influenced the layer specific remyelination pattern and whether sheaths were replaced or newly formed at a previously unmyelinated axon.

2) It would be informative to add a Y-projection of an example region from cuprizone-treated animals at a late time point of recovery, similar to Figure 2A.

3) In Figure 3A and B, it is not easy to see where in the 3D space the individual data points are located. Is there a different way to illustrate this?

4) Since new cells are inserted in between stable cells in control animals, I would expect nearest neighbor distance in Figure 3C to be significantly lower in "all cells" vs. "stable cells". Please comment.

5) Referring to Figure 4E and F, please add a description of how volumes for randomly appearing oligodendrocytes were predicted.

6) In the "2 neighbors" subpanel of Figure 6B, it appears like part of the right neighbor at 3 weeks of cuprizone and the left neighbor at 10d of recovery were lost, too. Please comment.

7) Figure 3—figure supplement 1B does not seem to match Figure 3—figure supplement 1A (sheath length is < 40 μm at 0 dpf, curve should be steeper between 0 and 8 dpf). Please verify.

Reviewer #3:

In this manuscript Orthmann-Murphy et al. use in vivo two photon imaging to study the spatial patterns of remyelination in the cortex following cuprizone intoxication, making several interesting findings. These include that oligodendrocytes are replaced in different locations to the developmentally born oligodendrocytes, and that they are relatively inefficiently replaced within deeper cortical layers. Where new oligodendrocytes have overlapping domains with the lost oligodendrocytes, the previously existing myelin internodes can be replaced with a surprising degree of fidelity, including replication of the previous locations of nodes of Ranvier. Maintained expression of some of the cytoskeletal components of the node by demyelinated axons may contribute to this. In spite of the ability of new oligodendrocytes to replicate many previously existing myelin segments, the overall pattern of myelin is substantially disrupted following de/remyelination; the authors provide an excellent discussion of the implications of this for circuit function in the remyelinated cortex.

Although there are some inherent limitations in the study (most notably that the patterns of remyelination seen in the cuprizone model may or may not extrapolate to other rodent models or to MS), the work is highly original and of a very high standard. The findings are novel and highlight the power of in vivo live imaging techniques. I am enthusiastic about the manuscript and have only one request for revised presentation of data.

1) Figure 5E would benefit from a more comprehensive set of images showing a region of cortex at baseline, peak demyelination and then remyelination (in addition to the pseudocolored analysis showing novel and replaced segments at the end of the experiment), thoroughly documenting examples of internodes that are lost and then replaced. Although the authors emphasize the overall disruption of the previously existing patterns of myelin following remyelination, the fact that over half of the myelin internodes that degenerated were subsequently replaced (Figure 5D) is actually fairly remarkable.

---

## [Author Response]

Reviewer #2:[…] The paper is very well-written and technically convincing. I recommend publication with only few suggestions for the revision:1) The Discussion is well written, but the possibility that axonal damage may have occurred during cuprizone treatment is not mentioned. This may, at least partly, have influenced the layer specific remyelination pattern and whether sheaths were replaced or newly formed at a previously unmyelinated axon.

We agree that axonal damage or axonal loss could contribute to the alteration in myelination patterns after oligodendrocyte regeneration and the reduced recovery observed in deeper layers of the cortex. Please see the response to point #3 for reviewer #1. We now include data indicating that there isn’t a significant reduction in neuronal processes in these gray matter regions and have expanded our discussion of this point in the Discussion section.

2) It would be informative to add a Y-projection of an example region from cuprizone-treated animals at a late time point of recovery, similar to Figure 2A.

We have replaced Figure 2A panels with Y-projections from a cuprizone-treated mouse at baseline and then later at five weeks of recovery.

3) In Figure 3A and B, it is not easy to see where in the 3D space the individual data points are located. Is there a different way to illustrate this?

We agree that it is difficult to assess the exact relative positions in this plot. This problem is inherent with illustrating dense 3D data in 2D. To improve visualization of these plots, we now include an animated version of Figure 4B in Video 4.

4) Since new cells are inserted in between stable cells in control animals, I would expect nearest neighbor distance in Figure 3C to be significantly lower in "all cells" vs. "stable cells". Please comment.

In this analysis, “Stable cells” refers to the comparison of the position of persistent (non-degenerating) cells at baseline versus the final imaging time-point, providing an indication of the amount of movement (wobble) that arises through actual cell displacement or registration artifact over eight weeks of imaging. To clarify this measurement, we have changed the term “Stable cells” to “Self-self” in the figure.

For “All cells", we determined the location of each regenerated cell at five weeks recovery and calculated the distance to the nearest oligodendrocyte that was present at baseline. In control, the nearest neighbor distance increases slightly (though not significantly), because the newly generated cells were not present at baseline and thus were compared to other oligodendrocytes. Note that these cells were often further away than the movement that surviving oligodendrocytes experienced over this time period (Self-self distance), resulting in a greater nearest neighbor distance.

We have now clarified this issue in the Results (subsection “Regeneration alters the pattern of myelination in the cortex”) and the Figure 3 legend.

5) Referring to Figure 4E and F, please add a description of how volumes for randomly appearing oligodendrocytes were predicted.

We have now added an additional description of the method used to generate random arrays (Materials and methods section).

6) In the "2 neighbors" subpanel of Figure 6B, it appears like part of the right neighbor at 3 weeks of cuprizone and the left neighbor at 10d of recovery were lost, too. Please comment.

The reviewer is correct in noting that multiple myelin sheaths were lost in this example. Here, two myelin sheaths along a single axon are fully contained within the field at baseline. One of these sheaths is pseudocolored magenta at Baseline and the NOR positions of all the sheaths are now indicated with yellow arrowheads. The axon is continuously myelinated in this area; as a result, each of these sheaths is bound by two neighboring sheaths that extend out of the imaging area. After three weeks in cuprizone, the sheath to the right of the magenta sheath (as well as its neighbor to the right) has disappeared. After 10 days of recovery, the magenta sheath (as well as its neighbor to the left) have disappeared and the sheaths that border to the right have been regenerated. By five weeks of recovery, all sheaths along this axon (in this region) have been regenerated. The replacement for the magenta colored sheath is pseudocolored in green at this time point.

We have added yellow arrowheads to the figure to denote each NOR present at baseline and then Recovery to better illustrate the boundaries of the individual myelin sheaths.

7) Figure 3—figure supplement 1B does not seem to match Figure 3—figure supplement 1A (sheath length is < 40 μm at 0 dpf, curve should be steeper between 0 and 8 dpf). Please verify.

The apparent discrepancy in length between A and B in Figure 3—figure supplement 1 is due to the 2D representation of the sheath, which extended slightly in the z-dimension. All length measurements were performed in 3D.

A note has been added to the figure legend to highlight this issue.

Reviewer #3:[…] I am enthusiastic about the manuscript and have only one request for revised presentation of data.1) Figure 5E would benefit from a more comprehensive set of images showing a region of cortex at baseline, peak demyelination and then remyelination (in addition to the pseudocolored analysis showing novel and replaced segments at the end of the experiment), thoroughly documenting examples of internodes that are lost and then replaced. Although the authors emphasize the overall disruption of the previously existing patterns of myelin following remyelination, the fact that over half of the myelin internodes that degenerated were subsequently replaced (Figure 5D) is actually fairly remarkable.

Because we used a limited duration of cuprizone, cell loss and regeneration occur simultaneously during the recovery period (see Figure 2C). As a result, 2D images at these time points are extremely complex, with surviving sheaths and regenerated sheaths occupying the same area. To overcome this problem, we have now added a series of z-series videos (Videos 5-7) in which sheaths with different fates (lost, novel, replaced) have been pseudocolored and are overlayed with the EGFP+ myelin sheaths for a given volume at baseline, at three days recovery (peak demyelination) and at five weeks of recovery. These videos make it possible to evaluate what happens to individual sheaths. We hope that this mode of data representation provides greater clarity.